# Ssu72 phosphatase is a conserved telomere replication terminator

Jose Miguel Escandell[1,*,†] (ID), Edison SM Carvalho[1,†], Maria Gallo-Fernandez[1], Clara C Reis[1], Samah Matmati[2], Inês Matias Luís[3], Isabel A Abreu[3], Stéphane Coulon[2] (ID) & Miguel Godinho Ferreira[1,4,**] (ID)

## Abstract

Telomeres, the protective ends of eukaryotic chromosomes, are replicated through concerted actions of conventional DNA polymerases and elongated by telomerase, but the regulation of this process is not fully understood. Telomere replication requires (Ctc1/Cdc13)-Stn1-Ten1, a telomeric ssDNA-binding complex homologous to RPA. Here, we show that the evolutionarily conserved phosphatase Ssu72 is responsible for terminating the cycle of telomere replication in fission yeast. Ssu72 controls the recruitment of Stn1 to telomeres by regulating Stn1 phosphorylation at Ser74, a residue located within its conserved OB-fold domain. Consequently, $ssu72\Delta$ mutants are defective in telomere replication and exhibit long 3′-ssDNA overhangs, indicative of defective lagging-strand DNA synthesis. We also show that hSSU72 regulates telomerase activation in human cells by controlling recruitment of hSTN1 to telomeres. These results reveal a previously unknown yet conserved role for the phosphatase SSU72, whereby this enzyme controls telomere homeostasis by activating lagging-strand DNA synthesis, thus terminating the cycle of telomere replication.

**Keywords** CST; fission yeast; lagging-strand synthesis; SSU72; telomere
**Subject Categories** DNA Replication, Repair & Recombination; Post-translational Modifications, Proteolysis & Proteomics
**The EMBO Journal (2019) 38: e100476**

## Introduction

Telomeres are protein–DNA complexes that form the ends of eukaryotic chromosomes (reviewed in Palm & de Lange, 2008). The predominant function of telomeres is to prevent the loss of genetic information and to inhibit DNA repair at chromosome ends, thus maintaining telomere protection and genome stability.

Loss of telomere regulation has been linked to two main hallmarks of cancer: replicative immortality and genome instability (Hanahan & Weinberg, 2011). However, telomeres face an additional challenge: DNA replication. Due to G-rich repetitive DNA sequences and protective structures, telomeres represent a natural obstacle for passing replication forks (Maestroni et al, 2017), and replication fork collapse can lead to the loss of whole telomere tracts. To counteract these effects, telomerase (Trt1 in S. pombe and TERT in mammals) is responsible for adding specific repetitive sequences to telomeres, compensating for the cell's inability to fully replicate chromosome ends (Greider & Blackburn, 1985). However, it is currently incompletely understood how telomerase activity is regulated and how the telomerase cycle is coupled to telomeric DNA replication. Intriguingly, several DNA replication proteins are required for proper telomere extension (Dahlen et al, 2003). Conversely, specific telomere components are themselves required for proper telomere replication and telomere length regulation (Miller et al, 2006; Sfeir et al, 2009), suggesting that there is a thin line separating telomere replication and telomere elongation by telomerase.

Using fission yeast, Chang et al (2013) proposed a dynamic model that demonstrates how telomere replication controls telomere length and how this is carried out by the telomere complex. The telomere dsDNA-binding components Taz1, Rap1, and Poz1 promote the recruitment of Polα-Primase to telomeres. Because shorter telomeres possess less Taz1/Rap1/Poz1, Polα-Primase recruitment and lagging-strand synthesis are delayed, leading to the accumulation of ssDNA at telomeres. This event results in the activation of the checkpoint kinase Rad3[ATR] and the subsequent phosphorylation of telomeric Ccq1-T93, a step required for telomerase activation in fission yeast. Thus, as a consequence of delayed Polα-Primase recruitment to short telomeres and the subsequent accumulation of ssDNA, Rad3[ATR] is transiently activated leading to telomerase recruitment and telomere elongation.

Another complex known as CST (Cdc13/Stn1/Ten1 in S. cerevisiae and CTC1/STN1/TEN1 in mammals) is known to control telomere replication. This complex is responsible for both 5′-ssDNA

1  Instituto Gulbenkian de Ciência, Oeiras, Portugal
2  Equipe Labellisée Ligue, CRCM, CNRS, Inserm, Institut Paoli-Calmettes, Aix-Marseille University, Marseille, France
3  Instituto de Tecnologia Química e Biológica António Xavier, Universidade Nova de Lisboa, Oeiras, Portugal
4  Institute for Research on Cancer and Aging of Nice (IRCAN), INSERM U1081 UMR7284, CNRS, Nice, France
   *Corresponding author. Tel: +351 214464511; E-nail: jplanells@igc.gulbenkian.pt
   **Corresponding author. Tel: +33 493377775; E-nail: Miguel-Godinho.FERREIRA@unice.fr
   †These authors contributed equally to this work

strand protection from nucleolytic degradation and recruitment of the Polα-primase complex to telomeres, thus promoting telomere lagging-strand DNA synthesis (Lin & Zakian, 1996; Grossi *et al*, 2004). Notably, CST is not only required to recruit Polα-primase but is also responsible for the switch from primase to polymerase activity, which is required for gap-less DNA replication (Lue *et al*, 2014). In humans, in addition to its role in telomere replication (Surovtseva *et al*, 2009), the CST complex also functions as a telomerase activity terminator (Chen *et al*, 2012) by inhibiting telomerase activity through primer confiscation and direct interaction with the POT1-TPP1 dimer. However, the mechanism regulating these CST functions remains unknown. In fission yeast, although STN1 and TEN1 homologs exist, no Cdc13/CTC1 homolog has been identified to date (Martín *et al*, 2007). Recent studies revealed that Stn1 is required for replication of telomeres and subtelomeres (Takikawa *et al*, 2017; Matmati *et al*, 2018), supporting the conserved role of fission yeast (C)ST in DNA replication.

In agreement with the replication model for telomere length regulation (Greider, 2016), the telomere-binding protein Rif1 was shown to regulate telomere DNA replication timing by recruiting Glc7 phosphatase to origins of replication and inhibiting Cdc7 activities in budding yeast (Hiraga *et al*, 2014; Mattarocci *et al*, 2014). Notably, this role is conserved in other organisms such as fission yeast (Davé *et al*, 2014) and human cells (Hiraga *et al*, 2017). Importantly, $rif1^+$ mutants display long telomeres; this effect is suggested to be a result of origin firing dysregulation (Greider, 2016). However, how telomere replication is terminated and how this is coupled with the regulation of telomerase activity remain unknown. Here, we report that the protein phosphatase Ssu72 displays a conserved role as a telomere replication terminator. Ssu72 was previously identified as an RNA polymerase II C-terminal domain phosphatase and is highly conserved from yeast to human (Krishnamurthy *et al*, 2004). In addition, Ssu72 functions as a cohesin-binding factor involved in sister-chromatid cohesion by counteracting phosphorylation of the cohesion complex subunit SA2 (Kim *et al*, 2010). In fission yeast, in addition to regulating RNA polymerase activity, Ssu72 has been shown to regulate chromosome condensation (Vanoosthuyse *et al*, 2014). However, none of the previous studies have noted deregulated telomere replication. Our data strongly support an unexpected role for Ssu72 in controlling lagging-strand synthesis through the regulation of Stn1 serine-74 phosphorylation, thus reducing telomeric ssDNA and inhibiting telomerase recruitment.

## Results

### Ssu72 is a negative regulator of telomere elongation

We carried out a genome-wide screen for regulators of telomere homeostasis in *S. pombe* using a commercially available whole-genome deletion library (*Bioneer* Corporation). This library allowed us to identify new non-essential genes involved in telomere homeostasis in fission yeast (Fig 1A). Of the genes identified from the screen, we selected the highly conserved phosphatase $ssu72^+$ (SPAC3G9.04) as the most promising candidate for further characterization. We generated a deletion mutant (*ssu72Δ*) as well as a point mutant devoid of phosphatase activity (*ssu72-C13S*) and found that these two mutants possess longer telomeres (Fig 1B).

Additionally, we found that Ssu72 localized to telomeres in a cell cycle-dependent manner. We performed cell cycle synchronization using a *cdc25-22* block-release method in a *ssu72-myc*-tagged strain and measured Ssu72 binding to telomeres by chromatin immuno-precipitation (ChIP). Cell cycle phases and synchronization efficiency were measured using the cell septation index. Interestingly, Ssu72-myc is recruited to telomeres in late S phase and declines later in the cell cycle (Fig 1C) and is recruited to telomeres at approximately the same time as the arrival of the lagging-strand machinery at chromosome ends (Chang *et al*, 2013).

*ssu72Δ* cells displayed increased (~1 Kb) telomere lengths compared to wild-type telomeres (~300 bp) (Fig 1B). We set out to understand the nature of telomere elongation in the *ssu72Δ* mutant background. To test whether the telomere elongation was dependent on telomerase, *trt1Δ* (deletion mutant for the catalytic subunit of telomerase) and *ssu72Δ* double heterozygous diploids were sporulated. Of the resulting tetrads, *trt1Δ ssu72Δ* double mutants were selected and streaked for several generations in order to facilitate telomere shortening in the absence of telomerase. While *ssu72Δ* mutant cells displayed long telomeres, *ssu72Δ trt1Δ* double mutant shortens telomeres (Fig 1D) in a passage-dependent manner. ChIP experiments consistently demonstrated an accumulation of Trt1-myc at *ssu72Δ* telomeres compared to *wt* cells (Fig 1E). Thus, the longer telomeres exhibited by *ssu72Δ* mutants were a consequence of telomerase deregulation.

Two independent studies (Moser *et al*, 2012; Yamazaki *et al*, 2012) showed that Ccq1 phosphorylation at Thr93 is required for telomerase-mediated telomere elongation in fission yeast. Using Western blot shift analysis, we observed that Ccq1 was hyperphosphorylated in *ssu72Δ* cells when we compared to those of *wt* strains (Fig 1F). To further confirm that telomere elongation was telomerase-dependent, we repeated the previous experiment using a phosphorylation-resistant mutant version of Ccq1 (Moser *et al*, 2012). We germinated a double heterozygous *ccq1-T93A/+ ssu72Δ/+* mutant and analyzed its progeny. As expected, *ccq1-T93A ssu72Δ* double mutants displayed a similar telomere-shortening rate to that of the *ccq1-T93A* single mutants (Appendix Fig S1). In agreement with these results, we further showed that telomere length in *ssu72Δ* mutants was dependent on Rad3, the kinase responsible for Ccq1-T93 phosphorylation (Appendix Fig S2A), and not dependent on the checkpoint kinase Chk1 (Appendix Fig S2B). In addition, *ssu72Δ rad51Δ* double mutants displayed similar telomere lengths to *ssu72Δ* single mutants (Appendix Fig S2C). Taken together, our results demonstrate that Ssu72 is a negative regulator of telomerase, possibly counteracting Rad3 activation and Ccq1 phosphorylation.

### Telomere length regulation by Ssu72 phosphatase is synergistic with Rif1

In fission yeast, the presence of telomeric ssDNA results in Rad3 activation and telomere elongation (Moser *et al*, 2012). Thus, we investigated whether *ssu72Δ* mutants accumulated telomeric ssDNA. We carried out in-gel hybridization assays using a C-rich probe to measure the accumulation of G-rich DNA at telomeres. Notably, the *ssu72Δ* mutant strain showed an almost sixfold increase in G-rich telomere sequences (Fig 2A). We observed that the accumulation of ssDNA at telomeres is increased in *ssu72Δ* mutants compared to *rif1Δ* mutants, though both strains have

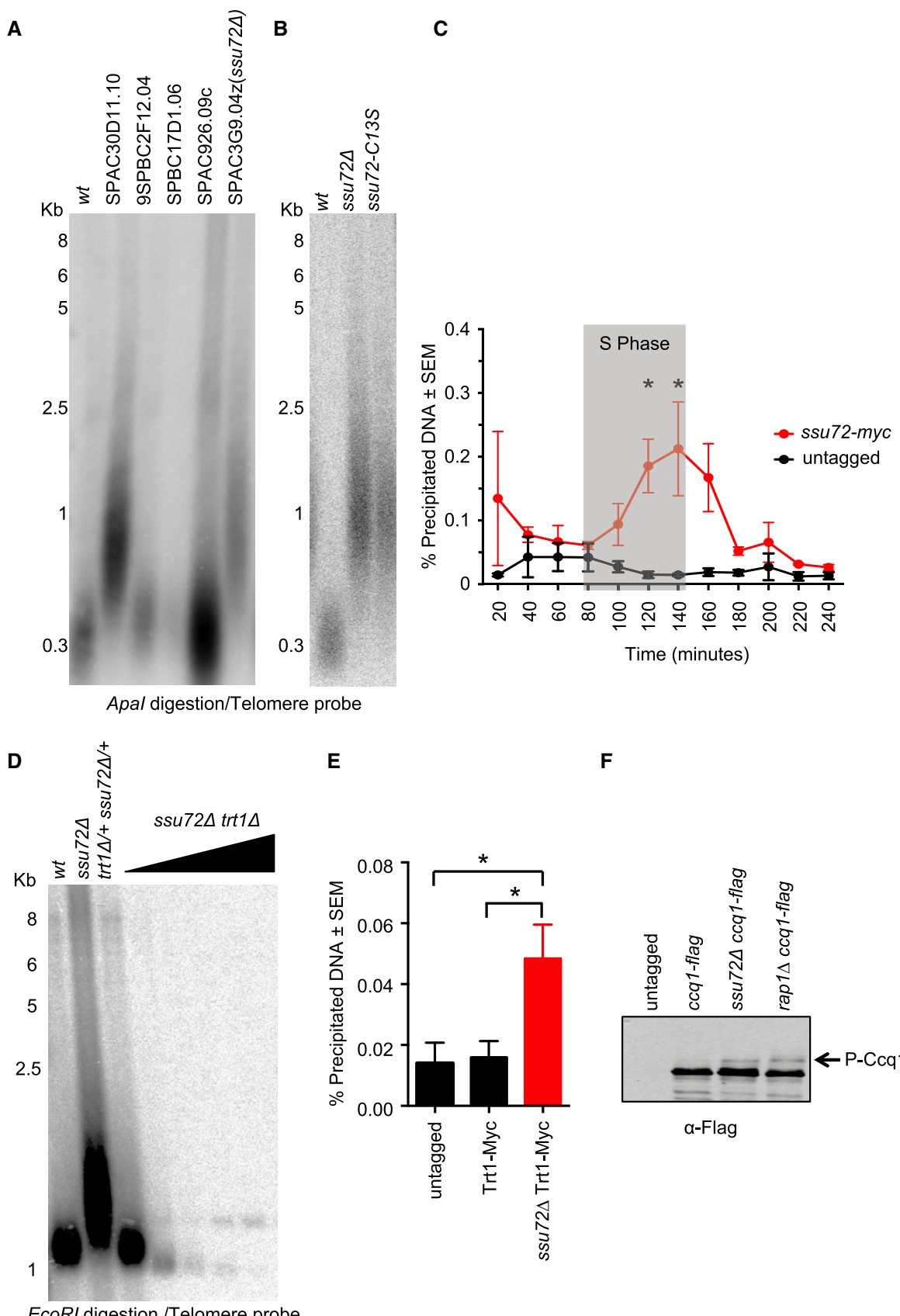

*ApaI* digestion/Telomere probe

*EcoRI* digestion /Telomere probe

α-Flag

**Figure 1.**

**Figure 1. Genetic screen identifies Ssu72 as telomerase regulator.**

A  We identified previously unknown telomere regulators in fission yeast using the haploid *S. pombe* whole-genome gene deletion library including *ssu72*[+] (SPAC3G9.04).

B  Telomere length of *wt*, *ssu72Δ*, and *ssu72-C13S* (point mutation at the phosphatase active site) strains were measured by Southern blot in *Apa*I-digested genomic DNA using a telomeric probe.

C  Ssu72 recruitment to telomeres is cell cycle regulated. Ssu72 was myc-tagged at the 3′ end, and ChIP analysis was carried out in cell cycle synchronized populations of *cdc25*[ts] strains. Septa formation was used proxy for S phase. $n \geq 3$; *$P \leq 0.05$ based on a two-tailed Student's *t*-test to *ssu72*[+] control samples. Error bars represent standard error of the mean (SEM).

D  Telomere length of *ssu72Δ* is dependent on telomerase. Diploid strains with the appropriate genotypes were sporulated, and *trt1Δ ssu72Δ* double mutants were streaked for multiple passages (triangle indicates increased number of generations).

E  Telomerase is recruited to telomeres in the absence of Ssu72. ChIP analysis for Trt1-myc in *wt* and *ssu72Δ* was performed as described in Materials and Methods using a non-tagged strain as a control. $n \geq 3$; *$P \leq 0.05$ based on a two-tailed Student's *t*-test to control sample. Error bars represent standard error of the mean (SEM).

F  Telomerase activator Ccq1 is hyperphosphorylated in *ssu72Δ* cells. *rap1Δ* cells were used as positive control for Ccq1 phosphorylation status. Western blots were performed using Ccq1-flag-tagged strains.

similar telomere lengths. To control for the observed increase of *ssu72Δ* telomeric ssDNA resulted from chromosome termini, we treated both *wt* and *ssu72Δ* DNA with exonuclease I (Appendix Fig S3). As expected, exonuclease I treatment reduced the overhang signal in *ssu72Δ*. In addition, we monitored Rad11[RPA]-GFP localization at telomeres in *ssu72Δ* mutant cells by live cell imaging and consistently detected an accumulation of this single-stranded binding protein at telomeres (Fig 2B). Thus, our data together suggest that *ssu72Δ* is defective in lagging-strand synthesis and possesses longer telomere overhangs.

Recently, the telomere-binding protein Rif1 was found to control DNA resection and origin firing by recruiting PP1A phosphatase to double-strand breaks and origins of replication (Zimmermann *et al*, 2013; Davé *et al*, 2014; Hiraga *et al*, 2014; Mattarocci *et al*, 2014). We wondered whether Rif1 was also responsible for recruiting the Ssu72 phosphatase to telomeres. To test this hypothesis, we combined *ssu72Δ* and *ssu72-C13S* (catalytically dead) mutants with *rif1Δ* and carried out of telomere length epistasis analyses. While single mutants displayed telomere lengths of 1 Kb, both *ssu72Δ rif1Δ* and *ssu72-C13S rif1Δ* double mutants displayed telomeres that were longer than 3 Kb (Fig 2C). Thus, although Rif1 and Ssu72 do not control telomere length via the same genetic pathway and Rif1 is therefore not responsible for Ssu72 recruitment to telomeres, the observed synergistic telomere length effect provides evidence for crosstalk between the two regulatory mechanisms.

## Ssu72 controls telomere length through the Stn1-Ten1 complex

CST constitutes a second highly conserved protein complex that regulates telomere length and telomerase activity. The budding yeast CST complex (Cdc13, Stn1, and Ten1) plays opposing roles at the telomeres. Cdc13 is required for telomerase recruitment and is activated through its interaction with Est1, a subunit of telomerase (Qi & Zakian, 2000). This interaction is promoted by the phosphorylation of Cdc13 at T308 by Cdk1(Cdc28). In contrast, the Siz1/2-mediated SUMOylation of Cdc13 at Lys908 promotes its interaction with Stn1 (Hang *et al*, 2011). This interaction is required, with Ten1, for polymerase alpha complex recruitment and telomere lagging-strand DNA synthesis (Grossi *et al*, 2004). However, the regulatory mechanism underlying these two opposite functions remains unknown. Despite the lack of Cdc13 homologs in fission yeast, the Stn1-Ten1 complex appears to play similar roles to those found in budding yeast and

mammals (reviewed in Giraud-Panis *et al*, 2010). Consequently, we hypothesized that Ssu72 controls telomere length through the Stn1-Ten1 complex. Because fission yeast *stn1*[+] and *ten1*[+] deletion mutants lose telomeres completely and survive only with circular chromosomes (Martín *et al*, 2007), we carried out our experiments in mutants carrying a hypomorphic *stn1-75* allele (Garg *et al*, 2014). Similar to *ssu72Δ* mutants, *stn1-75* mutants possess long telomeres (~1 Kb) (Fig 2D). In contrast to our previous genetic studies presented in Fig 2C, *stn1-75 ssu72Δ* double mutants displayed similar telomere lengths to those of single mutants. This result suggests that Ssu72 controls telomere length through the same pathway as the Stn1-Ten1 complex.

We then decided to investigate this genetic interaction using a different strategy. Fission yeast Stn1 is recruited to telomeres in a cell cycle-dependent manner (Moser *et al*, 2009a; Miyagawa *et al*, 2014), with peak telomere association in the S/G2 phases of the cell cycle. This coincides with Ssu72 recruitment to telomeres, as observed in our synchronization experiments (Fig 1C). Given the genetic association, we asked whether the recruitment of Stn1 to telomeres was dependent on the function of Ssu72. To test this hypothesis, we performed Stn1-myc ChIP experiments in *ssu72Δ* cells throughout the cell cycle (Fig 3A). As previously shown, Stn1-myc was recruited to telomeres in S/G2 cells (Moser & Nakamura, 2009). However, in the absence of Ssu72, the recruitment of Stn1 to telomeres was severely impaired (Fig 3A). Thus, our results show that Ssu72 function is required for cell cycle recruitment of Stn1 to telomeres.

Based on these findings, we asked whether DNA replication was perturbed at chromosome ends of *ssu72Δ* mutants. Genomic DNA derived from *wt* and *ssu72Δ* cells was isolated, subjected to NsiI digestion and analyzed on 2D gels. Chromosome ends were revealed by Southern blotting using a telomere-proximal STE1 probe (Miller *et al*, 2006; Audry *et al*, 2015). In the first dimension, we observed three distinct bands for the *wt* parental strain but corresponding longer smearing bands for *ssu72Δ* mutants. As expected, we observed Y structures derived from passing replication forks within the *Nsi*I fragment in *wt* cells (Fig 3B). In contrast, Y structures were not observed in *ssu72Δ* mutants denoting a reduced level of telomere DNA replication. In addition, DNA replication was 2–4 times lower at the rDNA locus in *ssu72Δ* mutants compared to wild-type cells (Fig 3B). These data are consistent with previous results for telomere and ribosomal DNA replication defects upon Stn1 inactivation (Takikawa *et al*, 2017), thus reinforcing the idea that Ssu72 controls Stn1 function throughout the genome.

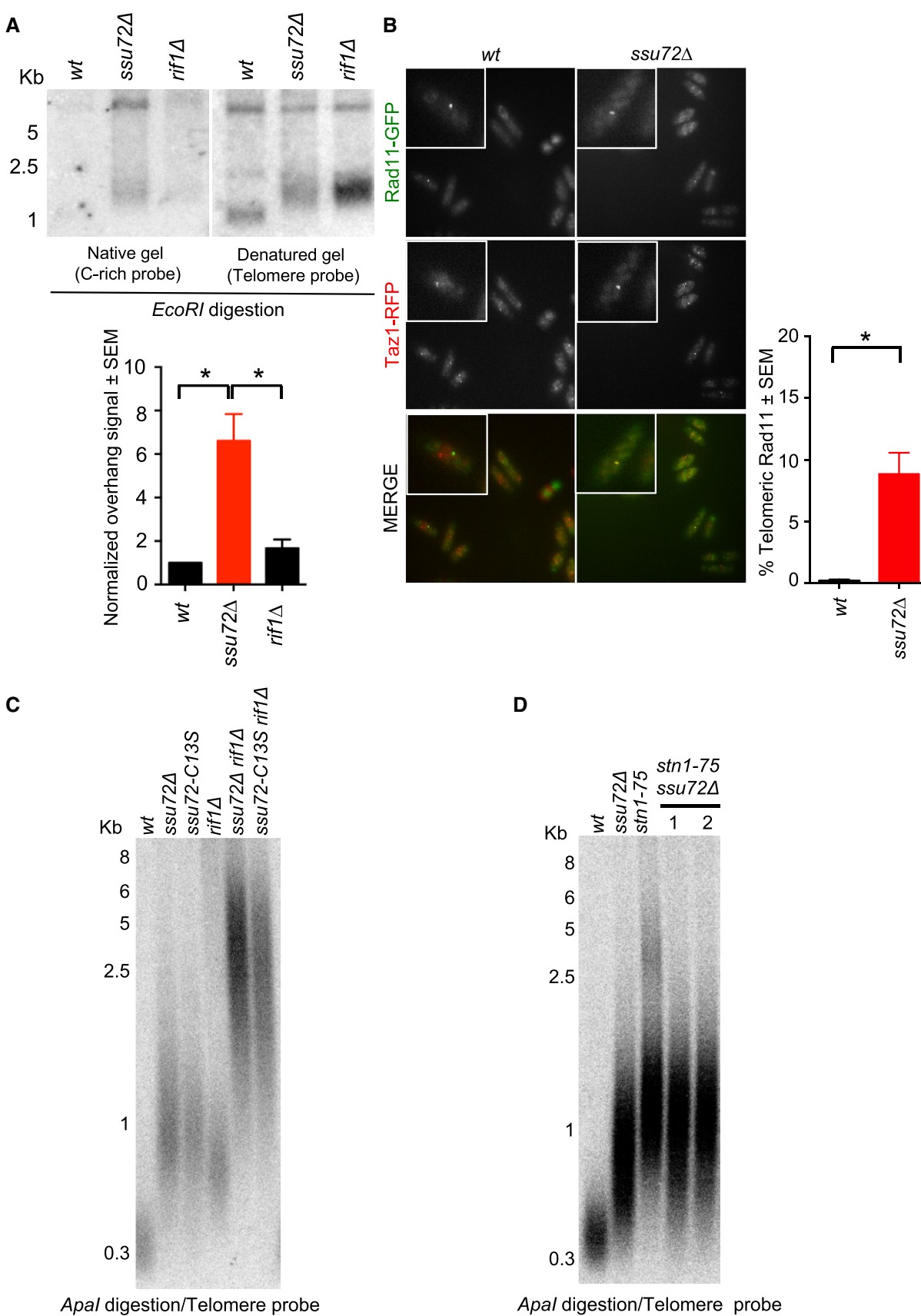

**Figure 2.**

**Figure 2. Ssu72 is required for telomeric C-strand stability.**

A   *ssu72Δ* telomeres present longer G-rich overhangs than *wt* and *rif1Δ*. In-gel hybridization in native and denaturing conditions was labeled with a radiolabeled C-rich telomere probe and quantified for ssDNA at the telomeres. *n* = 3; *P ≤ 0.05 based on a two-tailed Student's *t*-test to control sample. Error bars represent standard error of the mean (SEM).

B   RPA (Rad11-GFP) is enriched at *ssu72Δ* telomeres. Co-localization of Rad11-GFP with Taz1-mCherry, used as a telomere marker, was performed in *wt* and *ssu72Δ* cells; *n* = 3; *P ≤ 0.05 based on a two-tailed Student's *t*-test to control sample. Error bars represent standard error of the mean (SEM). More than 1,000 cells were analyzed for each genotype

C   *ssu72⁺* controls telomere length independently of *rif1⁺*. Epistasis analysis of telomere length of *ssu72Δ* and *ssu72-C13S* (catalytically inactive mutant) with *rif1Δ* was performed by Southern blotting of *ApaI*-digested DNA using a telomeric probe.

D   *ssu72⁺* and *stn1⁺* regulate telomere length in the same genetic pathway. Epistasis analysis of *ssu72Δ* and *stn1-75* performed by Southern blotting of *ApaI*-digested DNA using a telomeric probe. Two independently generated *ssu72Δ stn1-75* double mutants are shown.

## Ssu72 regulates Stn1 phosphorylation

Given that Ssu72 phosphatase activity is required to regulate telomere length and that Ssu72 is recruited to telomeres during the S/G2 phases, we hypothesized that Ssu72 might regulate Stn1 phosphorylation in a cell cycle-dependent manner. Previous studies have revealed different Cdk1-dependent phosphorylation sites in Stn1 in budding yeast (Liu *et al*, 2014; Gopalakrishnan *et al*, 2017). However, to date, Stn1 phosphorylation sites have not been identified in species outside of *S. cerevisiae*. Moreover, the phosphorylation sites described for budding yeast are not conserved in *S. pombe*. Thus, we decided to take an unbiased approach using mass spectrometry-based analysis of purified fission yeast Stn1. We immunoprecipitated Stn1-myc from cells carrying the *ssu72Δ* deletion. Subsequent analysis of the immunoprecipitated material revealed a phosphorylated peptide corresponding to Stn1 serine-74 (Fig EV1A). Notably, this serine is not only conserved in *Schizosaccharomyces sp* (the fission yeast genus) (Fig 3B) and budding yeast but also throughout higher eukaryotes, including humans and mice (Fig 3D). Therefore, we decided to mutate the serine-74 residue to aspartic acid (Stn1-S74D), a phosphomimetic amino acid.

Cells harboring the *stn1-S74D* mutation exhibited long telomeres (~1 Kb) similar to those found in *ssu72Δ* cells (Fig 3C). We hypothesized that telomere elongation in the *stn1-S74D* strain was telomerase-dependent. Consistent with this hypothesis, *stn1-S74D trt1Δ* double-mutant telomeres become shorter after sequential streaks (Fig EV2). Importantly, *stn1-S74D ssu72Δ* double mutants displayed similar telomere lengths to those in *stn1-S74D* single mutants (Fig 3C). In addition, we performed ChIP experiments in strains expressing Stn1-S74D-myc in order to analyze the recruitment of Stn1 to telomeres. Similar to our observations in mutants lacking *ssu72* phosphatase, Stn1-S74D-myc was not efficiently recruited to telomeres (Fig 3E). Taken together, our data suggest that fission yeast Stn1 is dephosphorylated at Ser74 to enable its efficient recruitment to telomeres and, consequently, efficient DNA replication and telomerase regulation.

Given that both Stn1 and Ssu72 are recruited to telomeres in S/G2 phases of the cell cycle and that Stn1-S74 dephosphorylation is required for efficient Stn1 recruitment to telomeres, we wondered whether Ssu72 phosphatase could counteract the action of a cell cycle-dependent kinase. Due to the central nature of Cdc2^Cdk1 in regulating the cell cycle, we mutated *ssu72⁺* in cells carrying the *cdc2-M68* temperature-sensitive mutant allele. At permissive temperatures (25°C), *ssu72Δ cdc2-M68* cells exhibited a similar telomere length to *ssu72Δ* single mutants (Fig EV3A). To inactivate Cdc2 activity, we grew *ssu72Δ cdc2-M68* strains at semi-permissive

temperatures. Our results show that progressive inactivation of Cdc2^Cdk1 in *ssu72Δ cdc2-M68* double mutants resulted in a decrease in telomere length at 32 and 34°C when compared to 19 and 25°C and with *ssu72Δ* single mutants. In order to have a more direct read-out for the involvement of Cdc2^Cdk1 on Stn1 regulation, we sought to test whether we could rescue Stn1 recruitment defects in *ssu72Δ* background by inactivating Cdc2^Cdk1 kinase activity. We took advantage of the recently published *cdc2-as-M17* allele (Aoi *et al*, 2014) that is sensitive to ATP analogs and is optimized to eliminate other physiological limitations. We constructed *ssu72Δ stn1-myc cdc2-as-M17,* and we treated either with DMSO or the ATP analog 1NM-PP1 for 3 h to inactivate cdc2^Cdk1 and measured Stn1 recruitment to telomeres by ChIP. While DMSO-treated *ssu72Δ stn1-myc cdc2-as-M17* cells showed lower Stn1 recruitment to telomeres, we were able to rescue the *ssu72Δ* defect by inactivating *cdc2-as* with ATP analog (Fig EV3B). Therefore, even though Stn1 serine-74 does not lie within a conserved CDK consensus site, our data indicate that Cdc2^Cdk1 activity counteracts Ssu72 phosphatase, perhaps indirectly via cell cycle regulation.

## Ssu72 is required for DNA polymerase α DNA replication

The CST complex is thought to perform different functions. In humans, it was proposed to be a terminator of telomerase activity due to its higher affinity for telomeric single-stranded DNA formed after telomerase activation (Chen *et al*, 2012). In addition, CST is thought to promote the restart of stalled replication forks (Gu *et al*, 2012; Stewart *et al*, 2012). In fission yeast, the (C)ST complex also exhibits this dual function. First, binding of this complex to telomeres inhibits telomerase function through the interaction of the SIM domain of Stn1 with K242-SUMOylated Tpz1 (Garg *et al*, 2014; Miyagawa *et al*, 2014; Matmati *et al*, 2018). Second, Stn1 participates in telomere and subtelomere semi-conservative DNA replication (Takikawa *et al*, 2017; Matmati *et al*, 2018).

Budding yeast CST promotes lagging-strand synthesis by interacting with the catalytic and B-subunits of DNA polymerase α-primase (Qi & Zakian, 2000; Chandra *et al*, 2001; Grossi *et al*, 2004). We thus hypothesized that Ssu72 could regulate DNA polymerase α-primase at fission yeast telomeres. To test this hypothesis, we carried out genetic epistasis analyses with the catalytic subunit of the polymerase α complex (*pol1⁺*). Hypomorphic mutations in polymerase α result in longer telomeres in fission yeast due to the formation of 3′ overhangs that sustain telomerase activation (Dahlen *et al*, 2003). As expected, while *pol1-13* had telomeres of approximately 1Kb in length, *pol1-13 ssu72Δ* double mutants had similar telomere lengths to single mutants (Fig 4A), suggesting they would

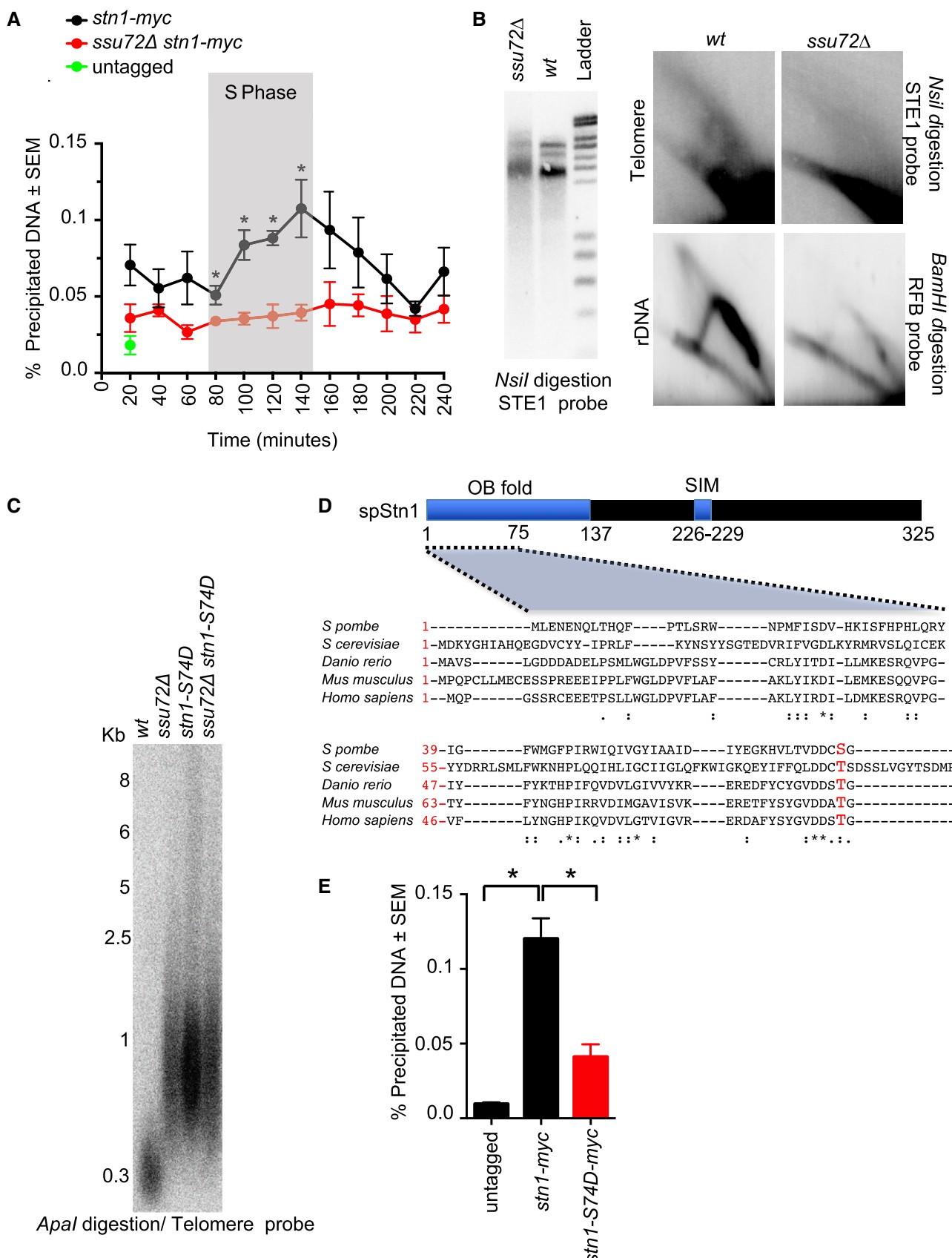

Figure 3.

◀

**Figure 3.  Ssu72 controls Stn1 telomere recruitment and phosphorylation.**

A   Ssu72 is required for telomere recruitment of Stn1 in late S phase. ChIP analysis of *stn1-myc* in *wt* and *ssu72Δ* cells was performed in synchronized *cdc25*[ts] cells. $n \geq 3$; *$P \leq 0.05$ based on a two-tailed Student's *t*-test to *ssu72*[+] control samples. Error bars represent standard error of the mean (SEM).

B   2D-gel analysis of *Nsi*I telomeric fragments of *wt* and *ssu72Δ* strains. Smart ladder from *Eurogentec* was used for DNA size measurement.

C   Serine-74 substitution to a phosphomimetic aspartate amino acid (*stn1-S74D*) is sufficient to confer *ssu72Δ* telomere defects. Telomere length epistasis analysis of *ssu72Δ* and *stn1-S74D* mutants was performed by Southern blotting of *Apa*I-digested genomic DNA using a telomeric probe.

D   Sequence alignment of Stn1 highlighting serine-74 identified in fission yeast as a phosphorylated residue.

E   Similar to *ssu72Δ* mutants, *stn1-S74D* is defective in telomere recruitment. ChIP analysis of *stn1-myc* and *stn1-S74D-myc* using a non-tagged strain as a control. $n = 3$; *$P \leq 0.05$ based on a two-tailed Student's *t*-test to control sample. Error bars represent standard error of the mean (SEM).

be involved in the same genetic pathway. In addition, we carried out similar epistasis studies with both RNA primase subunits (Spp1 and Spp2) and observed that *spp1-9 ssu72Δ* and *spp2-9 ssu72Δ* double mutants exhibited similar telomere lengths to those of single mutants (Appendix Fig S4A). Taken together, these results indicate that Ssu72 function at telomeres relies on the activity of both DNA polymerase α and RNA primase complexes. Thus, similarly to Stn1, Ssu72 controls lagging-strand synthesis at telomeres.

We next investigated whether Stn1 overexpression was sufficient to rescue the telomere defects observed in *ssu72Δ* mutants. To test this hypothesis, we replaced the *stn1*[+] endogenous promoter with inducible thiamine-regulated *nmt* promoters (Basi *et al*, 1993). We observed that none of the promoters used to overexpress Stn1 rescued the telomere defects of *ssu72Δ* (Appendix Fig S4B), indicating that Stn1 overexpression was unable to compensate for the defects in *ssu72Δ*.

CST in budding yeast regulates lagging-strand synthesis by stimulating DNA polymerase activity through the interaction of Stn1 with Pol1 (Lue *et al*, 2014). Similarly, we postulated that the mechanism whereby Ssu72 phosphatase controls lagging-strand synthesis is achieved by regulating the Stn1-Pol1 interaction. To test this hypothesis, we carried out immunoprecipitation experiments using extracts derived from Pol1-Flag and Stn1-Myc-tagged strains. As a control, we first verified that we could co-purify Ten1-Flag with Stn1-Myc in *ssu72Δ* mutants. In agreement, we could readily immunoprecipitate Ten1-Flag and Stn1-Myc in both the *wt* and *ssu72Δ* strains (Fig 4B). Thus, Ssu72 phosphatase does not regulate the Stn1-Ten1 interaction. Similar to what was previously observed in budding yeast, we were also able to reveal Pol1-Stn1 interaction in wt cells (Fig 4C). In contrast, using extracts derived from *ssu72Δ* cells, we were unable to immunoprecipitate Stn1-Myc with Pol1-Flag using anti-Flag antibodies. Our results show that Ssu72 is required for the interaction of Stn1 with DNA polymerase alpha, suggesting that Ssu72 functionality relies on Stn1-dependent activation of lagging-strand synthesis.

The results from the previous experiment suggest that Ssu72 is required to activate DNA polymerase α at telomeres. To test this, we overexpressed the catalytic subunit of polymerase α (*pol1*[+]) in cells lacking Ssu72. Previous studies showed that overexpression of *pol1*[+] was sufficient to rescue strains with lagging-strand synthesis defects (Dahlen *et al*, 2003). Remarkably, using *pol1*[+] expression from multicopy plasmids, we showed that overexpression of *pol1*[+] in *ssu72Δ* mutants partially rescued telomere defects of *ssu72Δ* mutants (Fig 4D).

Altogether, our results suggest that Ssu72 phosphatase regulates Stn1 phosphorylation status in order to control Stn1 recruitment to telomeres and DNA polymerase α activation of lagging-strand synthesis. We propose a dynamic model where Rif1-dependent

phosphatase activities regulate telomere replication initiation by controlling origin firing. Further, Ssu72 phosphatase functions as a telomere replication terminator by regulating Stn1 recruitment to telomeres in a cell cycle-dependent manner to activate lagging-strand synthesis, thus ending the telomere replication cycle (Fig 4E).

## SSU72 telomere function is conserved throughout evolution

Because Ssu72 is a highly conserved phosphatase and CST has similar functions in different species, we tested whether telomere regulation by SSU72 was conserved in human cells. In contrast to fission yeast, *SSU72* is an essential gene in both budding yeast (Ganem *et al*, 2003; Krishnamurthy *et al*, 2004) and mice (Kim *et al*, 2016). Consistently, we were unable to produce human cell lines lacking SSU72 using conventional CRISPR/Cas9 technology, suggesting that SSU72 is essential in humans. Therefore, we resorted to use short hairpin RNAs to downregulate SSU72 protein levels. This approach has been previously used in human cells to study the role of SSU72 in mammals (Kim *et al*, 2010).

Our results show that, similar to fission yeast, knockdown of SSU72 in human cells causes telomere dysfunction. We downregulated SSU72 levels using two specific shRNAs and collected HT1080 cells for analysis of telomere length 6 weeks after infection. In HT1080 cells, the median telomere length is 3.4 Kb in cells transfected with a control luciferase shRNA (Fig 5A and Appendix Fig S5A). As observed in fission yeast, downregulation of SSU72 using shRNAs against the protein-coding sequence (CDS, knockdown efficiency 85%) or the untranslated region (UTR, knockdown efficiency 92%) results in an increase in median telomere length to 3.7 and 3.8 Kb, respectively (Fig 5A and Appendix Fig S5). Next, we hypothesized that the observed telomere elongation results from telomerase activity. To test this, we used the telomerase inhibitor BIBR1532. Treatment of cells with BIBR1532 resulted not only in the inhibition of telomere elongation in shSSU72-infected cells but also in a general decrease in telomere length in all treated cells (Fig 5A). In addition, telomere elongation in SSU72 shRNA-treated cells is accompanied by ssDNA accumulation at the telomere terminus (Appendix Fig S6). Thus, as in fission yeast, SSU72 controls overhang formation and telomere length by regulating telomerase function in human cells.

Human STN1 regulates telomere length by inhibiting telomerase activity (Chen *et al*, 2012). We decided to investigate whether human SSU72 and STN1 control telomere length through the same genetic pathway. To test this, we treated HT1080 cells for 4 weeks with control GFP shRNA, single SSU72 and STN1 shRNA, and double SSU72/STN1 shRNA, and measured telomere length using Southern blotting. Consistent with our fission yeast epistasis

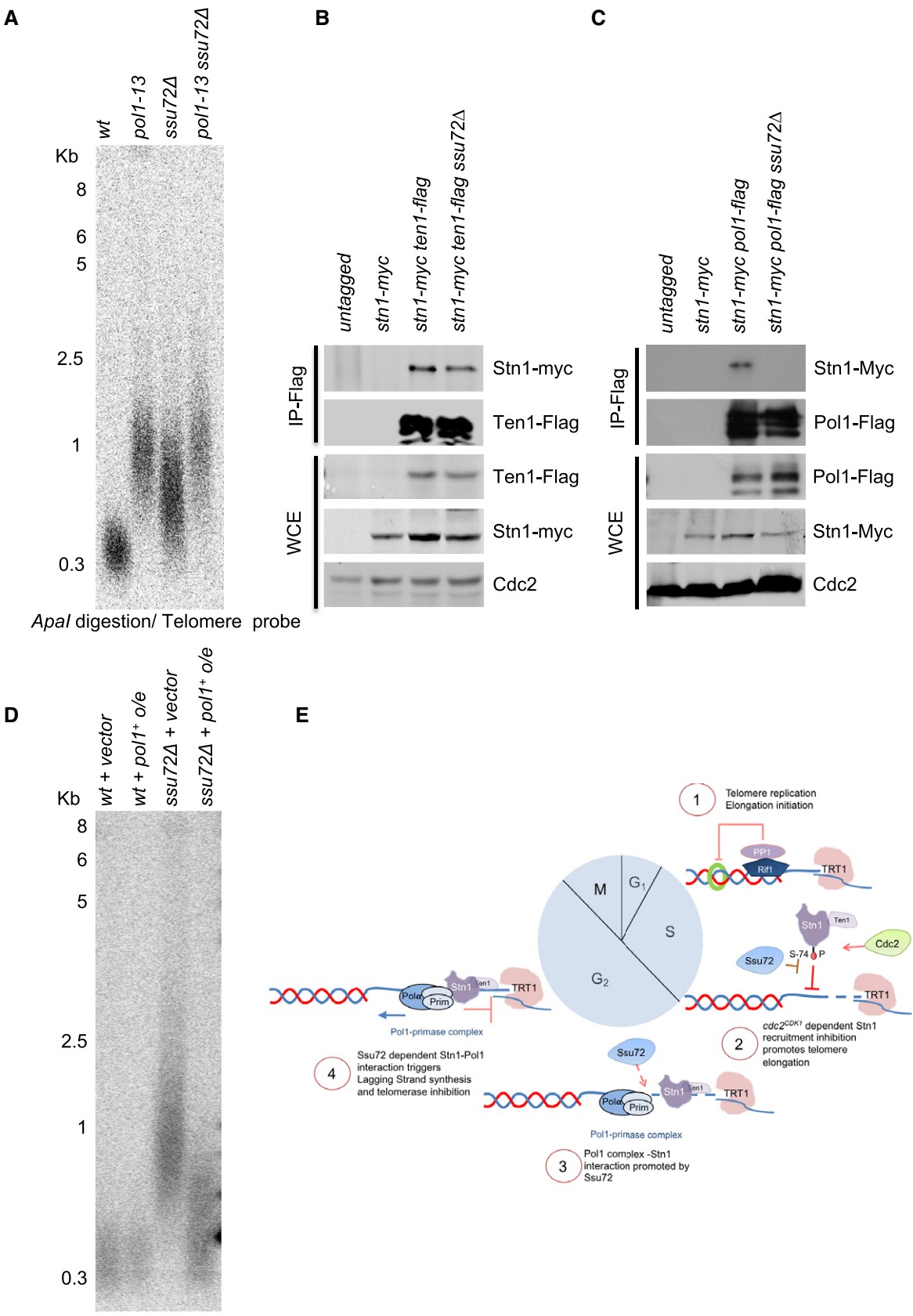

**Figure 4.**

◄

**Figure 4.  Ssu72 is required for polymerase α activation.**

A    $ssu72^+$ and $pol1^+$ regulate telomere length in the same genetic pathway. Epistasis analysis of $ssu72\Delta$ and $pol1\text{-}13$ mutants was performed by Southern blotting of ApaI-digested DNA using a telomeric probe.

B    Stn1-Pol1 interaction requires Ssu72 phosphatase. Immunoprecipitation experiments of Pol1-Flag with Stn1-Myc were performed in both wt and $ssu72\Delta$ mutants.

C    Immunoprecipitation of Ten1-Flag with Stn1-Myc in either wt or $ssu72\Delta$ was performed as control for Stn1-Ten1 complex integrity.

D    Overexpression of polymerase alpha partially rescues telomere length defect of $ssu72\Delta$ mutants. Multicopy vector with polymerase α under thiamine promoter was expressed in both wt and $ssu72\Delta$ cells.

E    Model for Ssu72 regulation of telomere replication in fission yeast. See text for details.

analysis, while single SSU72 and STN1 shRNA-treated cells showed telomere elongation, double shRNA-treated cells exhibited similar telomere elongation as the single shRNA treatments (Fig 5B and Appendix Fig S7). Thus, our data imply that SSU72 and STN1 control telomere length through same genetic pathway in human cells.

We next asked whether SSU72 downregulation results in DNA replication defects at human telomeres. As in previous studies, we used the appearance of multi-telomeric signals (MTS) as a readout for faulty DNA replication at telomeres (Sfeir et al, 2009). Following shRNA treatment, we measured MTS in metaphase spreads of HT1080 cells and observed that, while 9.3% of telomeres showed MTS in control shLuciferase-treated cells, SSU72 downregulation using either CDS or UTR shRNAs resulted in higher MTS levels (13.2 and 14.18%, respectively, $P \leq 0.01$) (Fig 5C). As control, we used aphidicolin, a DNA polymerase inhibitor shown to induce high levels of MTS in mammalian cells (Sfeir et al, 2009). Indeed, aphidi-colin treatment of HT1080 cells resulted in elevated levels of MTS in control luciferase shRNA-treated cells ($P \leq 0.0001$). However, aphidicolin did not result in a further increase of MTS in cells where SSU72 had been downregulated using shRNAs against either CDS or UTR ($P = 0.6$ and $P = 0.16$, respectively) (Fig 5C). This result suggests that higher MTS levels observed in SSU72 downregulated cells are a consequence of collapsing replication forks, thus high-lighting a role of SSU72 in controlling DNA replication in human cells. As expected, SSU72 downregulation in HT1080 cells also resulted in telomere-induced foci (TIF), as measured by the localiza-tion of 53BP1 to telomeres (Fig 6A). Importantly, TIF formation is not cell line dependent, as we also observed TIFs in HeLa cells treated with an siRNA against SSU72 (Fig EV4A).

Consistent with previous results, STN1 downregulation in human cells also resulted in increased MTS levels (Stewart et al, 2012). Likewise, STN1 dysfunction does not lead to higher levels of MTS upon aphidicolin treatment, mirroring to our findings in SSU72 downregulated cells. Moreover, we observed that the increase in MTS levels in SSU72 downregulated cells does not depend on telom-erase activity. SSU72 downregulation is still able to induce higher MTS levels in telomerase-negative U2OS cells (Fig EV4B). These

data are consistent with STN1 dysfunction, as downregulation of this factor in U2OS cells induces equivalent rates of replication fork stalling at telomeres (Stewart et al, 2012).

Our data suggest that the downregulation of SSU72 in human cells mimics previous results obtained in STN1 downregulated cells. Thus, we tested whether cells lacking SSU72 were defective for STN1 recruitment to telomeres. We expressed FLAG-tagged STN1 in HT1080 cells and infected these cells with lentiviral particles expressing an shRNA against either SSU72 or luciferase. We then performed telomeric ChIP experiments using FLAG antibodies (Fig 6B). Upon downregulation of SSU72, we observed a 40% reduction in STN1 binding to telomeres compared to shLuciferase-treated cells. Together, our data reveal an evolutionarily conserved role for SSU72 phosphatase in controlling STN1 recruitment to telomeres and, therefore, in regulating lagging-strand syntheses at telomeres.

# Discussion

Protein phosphorylation plays key regulatory roles in almost all aspects of cell biology. Even though the function of protein kinases in telomere biology has been widely studied, the role of phos-phatases remains mostly unexplored. Contrary to this trend, budding yeast Pph22 phosphatase was recently shown to regulate the phosphorylation of Cdc13 in a cell cycle-dependent manner (Shen et al, 2014). The dephosphorylation of specific Cdc13 resi-dues by Pph22 reverses the interaction of Cdc13 with Est1 and, consequently, disengages telomerase from telomeres (Shen et al, 2014).

However, to date, there are no known phosphatases that regulate telomere length in fission yeast or higher eukaryotes. The data presented here depict an unprecedented role for a highly conserved phosphatase in telomere regulation. Ssu72 belongs to the group of class II cysteine-based phosphatases, homologous to low molecular weight phosphatases, and some bacterial arsenate reductases (Alonso & Pulido, 2016). Although best known for its role as an RNA polymerase II CTD phosphatase in diverse species

**Figure 5.  Downregulation of human SSU72 results in telomere elongation and fragility.**

A    Telomere elongation of SSU72 downregulated cells is telomerase-dependent. Telomere Southern blot was carried out with HT1080 cells infected for 4 weeks with lentiviral particles carrying two independent shRNAs against SSU72 (CDS and UTR regions) and luciferase (Luc) as control for shRNA. Knockdown efficiencies were determined by RT–qPCR using specific primers against hSSU72. Densitometry analysis is shown in Appendix Fig S5.

B    Telomere elongation of SSU72 downregulated cells is epistatic with STN1 downregulation. TRF analysis of HT1080 cells infected with the appropriate retroviral or lentiviral particles carrying either GFP shRNA, SSU72 shRNA (against UTR), STN1 shRNA, or double SSU72 shRNA/STN1 shRNA. Cells were grown for 4 weeks, and DNA was isolated to carry out Southern blot analysis. Densitometry analysis is shown in Appendix Fig S7.

C    hSSU72 downregulation results in multi-telomeric signals (MTS) that are dependent on DNA replication. Visualization of mitotic spreads of HT1080 hSSU72 shRNA cells treated with aphidicolin and colcemid. Telomere FISH was carried out using a PNA-telomere probe. MTS are represented by arrows. Quantification of MTS per metaphase: $n = 4$; **$P \leq 0.01$ ****$P \leq 0.0001$ based on a two-tailed Student's t-test to control sample. Error bars represent standard error of the mean (SEM).

►

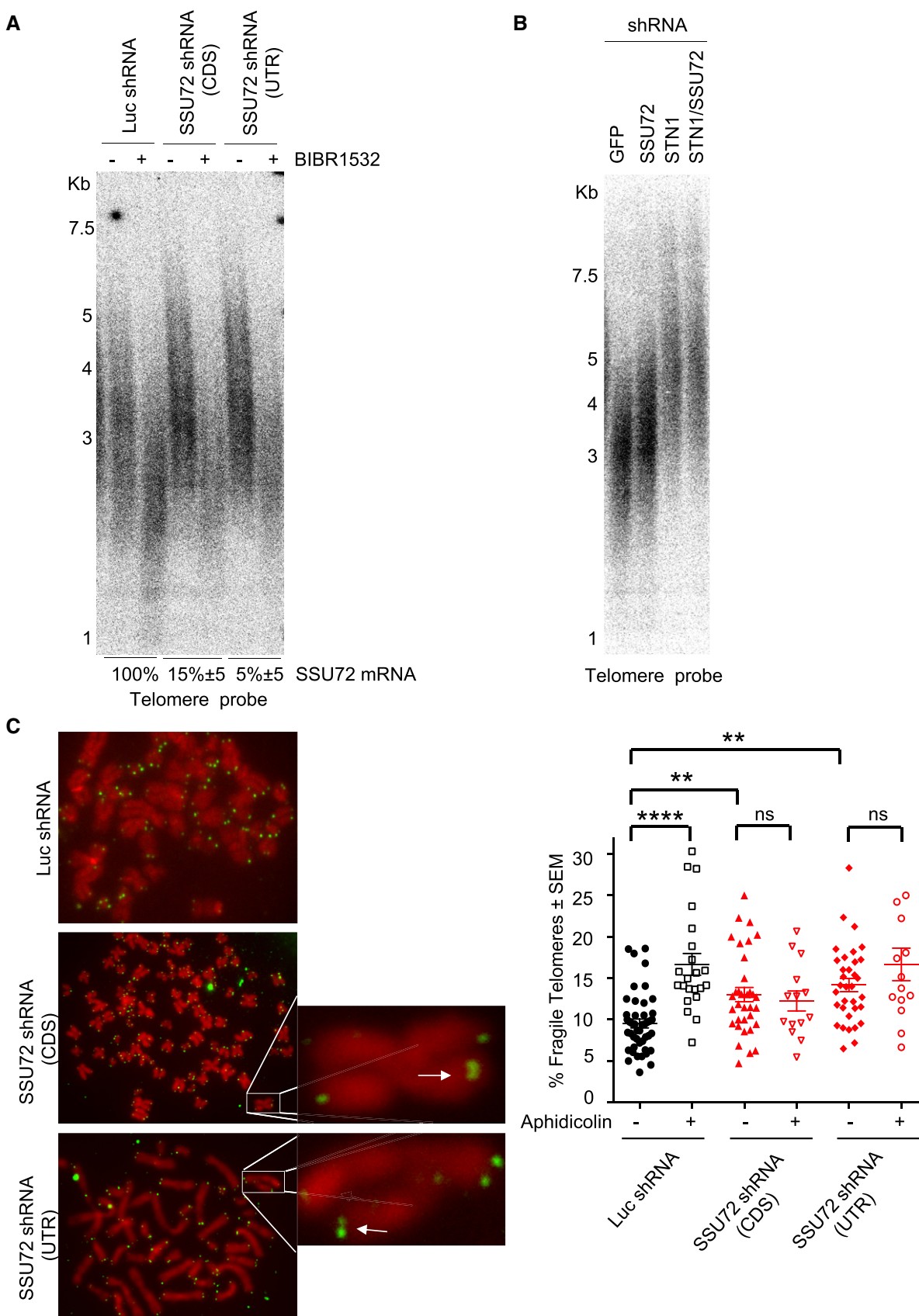

**Figure 5.**

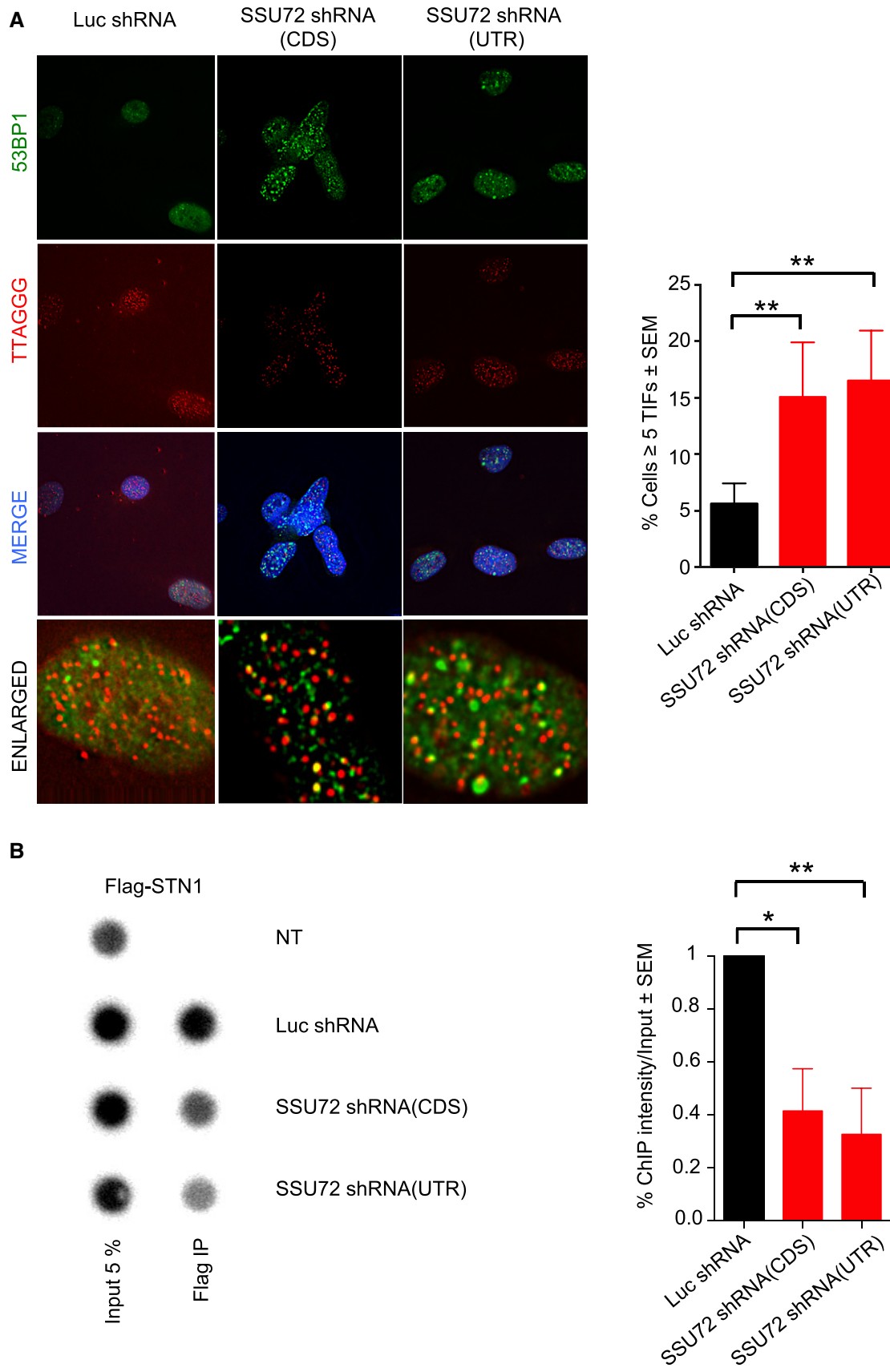

**Figure 6.**

◄

**Figure 6.  SSU72 is required for STN1 recruitment to human telomeres.**

A  hSSU72 downregulation results in telomere DNA damage foci (TIF). Cells with indicated shRNAs were fixed, and IF-FISH analysis was performed using a 53BP1 antibody and a PNA-telomere probe. (Right) Quantification of cells with more than 5 telomeric 53BP1 foci: $n = 3$; $**P \leq 0.01$ based on a two-tailed Student's $t$-test to control sample. Error bars represent standard error of the mean (SEM).

B  hSSU72 is required for efficient loading of hSTN1 at human telomeres. ChIP was performed using anti-FLAG antibodies, and Southern blotting was carried out using a human telomeric probe. Quantification of 3 independent ChIP experiments $*P \leq 0.05$ $**P \leq 0.01$ based on a two-tailed Student's $t$-test to control sample. Error bars represent standard error of the mean (SEM).

(Krishnamurthy *et al*, 2004), human SSU72 has also been identified as a protein that can interact with the tumor suppressor retinoblastoma protein (St-Pierre *et al*, 2005) and can target STAT3 signaling and Th17 activation in autoimmune arthritis (Lee *et al*, 2017). Other functions have been reported in human cells, including cohesin SA2 dephosphorylation (Kim *et al*, 2010). Consistent with the multiple roles for protein phosphatases, our work demonstrates that Ssu72 phosphatase also regulates rDNA and telomere replication, by controlling recruitment of Stn1 and promoting the Stn1-Pol1 interaction, thus activating lagging-strand DNA synthesis.

At present, it is not well understood how lagging-strand DNA replication inhibits telomerase activity. In fission yeast, Rad3 kinase is activated by the generation of ssDNA during DNA replication, leading to telomerase recruitment through the phosphorylation of Ccq1 at Thr93 (Moser *et al*, 2012). In this model, fill-in reactions by lagging-strand polymerases reduce ssDNA at telomeres, thus contributing to a negative feedback loop. Consistent with the role of telomeric ssDNA in activating telomerase in fission yeast, *ssu72Δ* cells have longer overhangs, extensive phosphorylation of Ccq1, and higher levels of telomerase at telomeres. In addition, Rad3 and Ccq1-T93 phosphorylation are both required for the elongation of telomeres in *ssu72Δ* mutants. Therefore, it is still possible that Ssu72 phosphatase activity is also required to regulate Ccq1 phosphorylation. Further experiments will be required to test this hypothesis.

Our work revealed that phosphorylation of Stn1 at Ser74 is required for the regulation of telomere length. Mutation of Stn1 Ser74 to aspartic acid, an amino acid that mimics constitutive phosphorylation, results in telomeric elongation that is epistatic with the *ssu72Δ* mutation. This result indicates that Ser74 phosphorylation is sufficient to explain the regulation of telomere length by Ssu72. The phosphorylation site identified in our mass spectrometry analysis resides within the OB-fold domain of Stn1. OB-fold phosphorylation is known to regulate protein–DNA binding and protein–protein interactions. For example, phosphorylation of human TPP1 (Tpz1 ortholog) in the OB-fold domain regulates the telomerase-TPP1 interaction (Zhang *et al*, 2013). In fission yeast, our data suggest that Stn1 phosphorylation at Ser74 may prevent its binding to telomeric DNA in early S phase. In addition, our data showed that $cdc2^{CDK1}$ activity inhibits Stn1 recruitment to telomeres in *ssu72Δ* background. Moreover, and in contrast to other cell cycle kinases such as $hsk1^{CDC7}$ and $plo1^{PLK}$, telomere elongation of *ssu72Δ* mutants is reversed with increased inactivation of the *cdc2-M68* temperature-sensitive allele. Interestingly, Ssu72 is recruited to telomeres in the S/$G_2$ phases concomitantly with the lagging-strand synthesis polymerases (Moser *et al*, 2009b). Although further experiments are required, an attractive model involves $Cdc2^{CDK1}$ phosphorylation of Stn1, thus creating a delay in lagging-strand synthesis and allowing telomere elongation. However, we cannot rule out that the rescue of Stn1 recruitment to telomeres achieved by Cdc2 inactivation in *ssu72Δ* mutants is indirect, perhaps working via a cell cycle-dependent event. Further, Ssu72 phosphatase reverses this process by promoting Stn1 binding to DNA polymerase α (Fig 4E). Notably, dephosphorylation of Stn1 has to be coordinated with Tpz1 SUMOylation, which is crucial for the recruitment of Stn1 to telomeres. Thus, both dephosphorylation and SUMOylation-mediated interactions with Tpz1 control the recruitment and activity of telomeric Stn1-Ten1.

The SSU72 phosphatase appears to be conserved throughout evolution. Depletion of a human SSU72 homolog in the HT1080 cell line results in similar phenotypes to those observed in fission yeast *ssu72Δ* mutants. First, SSU72 downregulation in HT1080 cells triggers DNA damage signaling at telomeres. Second, SSU72 depletion results in telomerase-dependent telomere elongation. Third, the recruitment of STN1 to telomeres is defective in SSU72-depleted HT1080 cells. Consistently, we observed increased telomere fragility (MTS) in SSU72-depleted cells, a phenotype also observed in STN1-deficient cells (Chen *et al*, 2012). We propose that SSU72 regulates STN1 recruitment to human telomeres in a manner similar to that in fission yeast. Currently, there is no evidence of STN1 phosphorylation in human cells. Nevertheless, Ser74 is conserved in humans as an amino acid capable of being phosphorylated (Thr81). Further analysis will determine whether human STN1 is phosphorylated at this residue and whether this modification regulates human telomere replication.

Recently, a model was proposed in which the replication fork regulates telomerase activity (Greider, 2016). This model describes how the regulation of origin firing and passage of the replication fork affect telomere homeostasis. In addition, we propose that telomere replication is controlled by two sets of phosphatases. On the one hand, the onset of telomere replication is regulated by Rif1-PP1A phosphatase through the inhibition of DDK activity at subtelomeric origins of replication. On the other hand, we now show that telomere lagging-strand synthesis is regulated by Ssu72 phosphatase, which promotes the Stn1-polymerase alpha interaction, thus terminating telomere replication and resulting in telomerase inhibition.

# Materials and Methods

### Yeast strains and media

The strains used in this work are listed in Appendix Table S1. Strains were constructed using commonly used techniques (Bähler *et al*, 1998). Standard media and growth conditions were used throughout this work (Martin *et al* 2007). For the strains containing pREP41 plasmids, cultures were grown overnight in media PMG (Pombe Glutamate Medium) with the required amino acids. To generate the *ssu72-C13S*, ssu72⁺ gene was cloned into pGEM vector using genomic DNA. pGEM vector was mutagenized to

create the pGEM-Ssu72-C13S. Endogenous ssu72$^+$ gene was deleted using a ura4$^+$ fragment, and FOA plates were used to select for Ssu72-C13S recombinants. Colonies were then screened for proper integration and sequenced to verify the presence of the point mutation. For *stn1-S74D* mutant strain, a genomic DNA fragment containing the *stn1$^+$* gene was cloned into pJK210 plasmid and mutagenized to create the pJK210 *stn1-S74D*. The *PacI*-linearized pJK210 *stn1-S74D* plasmid was transformed into wild-type strain, and cells were plated on minimal medium lacking uracil. Ura4-positive cells were streaked on FOA-plate to select for direct-repeat recombination between *stn1$^+$* and *stn1-S74D* allele. Presence of the *stn1-S74D* allele was subsequently verified by genomic sequencing.

### Southern blot analysis

Fission Yeast: Genomic DNA was obtained from exponentially growing yeast cells by phenol–chloroform extraction method. Human genomic DNA was extracted as described in Sfeir *et al* (2009). Approximately 2 μg of digested *ApaI* or *EcoRI* DNA in fission yeast or *AluI* and *MboI* for human cells was run in either 1% (fission yeast) or 0.6% (human cells) agarose gels. The gel was transferred to a positively charged nylon membrane, and telomere analysis was performed as described (Rog *et al*, 2009) or (Sfeir *et al*, 2009).

### Chromatin immunoprecipitation (ChIP)

In fission yeast, ChIP was performed as described (Moser *et al*, 2009a,b). Briefly, exponentially growing cells were fixed with 1% formaldehyde, 0.1M NaCl, 1 mM EDTA, 50 mM HEPES-KOH, and pH 7.5, and incubated for 20 min at room temperature. Then, the solution was quenched with 0.25 M glycine (final concentration) for 5 min. After 2 washes with cold PBS, cells were lysed with 2× lysis buffer (100 mM HEPES-KOH pH 7.5; 2 mM EDTA; 2% Triton X-100 0.2% Na deoxycholate) and disrupted by mechanical method. Chromatin was sheared, and equal amounts of DNA were used for immunoprecipitation protocol with either anti-Myc (9E10; Santa Cruz Biotechnology) or anti-Flag (M2-F4802; Sigma) with magnetic Protein A beads. After washing the DNA-protein complexes with 1st (lysis buffer/0.1% SDS/275 mM NaCl), 2nd (lysis buffer/0.1% SDS/500 mM NaCl), 3rd (10 mM Tris–HCl, pH 8.0, 0.25 M LiCl, 1 mM EDTA, 0.5% NP-40, 0.5% Na deoxycholate), recovered DNA by 50 mM Tris–HCl, pH 7.5, 10 mM EDTA, and 1% SDS was decroslinked, purified, and analyzed in triplicate by SYBR-Green-based real-time PCR (Bio-Rad) using the primers described in Carneiro *et al* (2010).

For human ChIP, cells were fixed in 1% formaldehyde in PBS and incubated for 15 min at room temperature. After quenching with glycine, cells were lysed with 1% SDS, 50 mM Tris–HCl pH 8.0, and 10 mM EDTA. After sonicating the chromatin, the diluted DNA-protein complexes were incubated with Flag magnetic beads (Sigma M8823) overnight. After 3 consecutive washes, with 1× in IP buffer (20 mM Tris pH 8, 0.15 M NaCl, 1% Triton X-100, 2 mM EDTA) with 0.1% SDS, 1× in IP buffer with 0.1% SDS, 0.5 M NaCl, and 10 mM Tris pH 8.0, 1 mM EDTA with 1% Nonidet, 1% Na deoxycholate, and 0.25 M LiCl. Complexes were eluted with 50 mM Tris pH 8.0, 10 mM EDTA, and 1% SDS. Decrosslinked DNA was denatured and slot-blotted into a Hybond N$^+$ membrane using a Bio-Rad blotter. Southern blot using a human telomere probe was carried out as described before (Sfeir & de Lange, 2012).

### Immunoblotting

Whole-cell extracts prepared using exponentially growing yeast cells were processed for Western blotting as previously described (Rog *et al*, 2009). Common Western blot techniques were used to detect different proteins. For detection of Myc-tagged proteins, we used anti-Myc monoclonal antibody (9E10; Santa Cruz) or rabbit anti-Myc (Abcam). For detection of Flag-tagged proteins, we used a Flag–M2 antibody (SIGMA—F1804). Cdc2 antibody (sc-53; Santa Cruz) was used as loading control.

### Immunoprecipitation

Exponentially growing yeast cells were lysed with IP Buffer x2 (50 mM HEPES pH 7.5; 150 mM NaCl; 40 mM EDTA; Triton X-100 0.5%; 0.1% NP-40; 0.5 mM Na$_3$VO$_4$; 1 mM NaF; 2 mM PMSF, 2 mM benzamidine; 10% glycerol; Complete proteinase inhibitor + DNase 10 U/ml). Equal amount of proteins was incubated with Flag–M2 (SIGMA - F1804) overnight followed by three washes for 10 min with IP buffer + 0.5 M NaCl. Common Western blot techniques were then used to detect proteins.

### Mass spectrometry analysis

Stn1 protein was purified by tagging C-terminus with 13-myc tag. 5 l of logarithmic cycling cells were collected and lysed with 50 mM HEPES [pH 7.5], 150 mM NaCl, 40 mM EDTA, 0.2% Triton X-100, 0.1% NP-40, 0.5 mM Na$_3$VO$_4$, 1 mM NaF, 2 mM PMSF, 2 mM benzamidine, 10% glycerol, DNase I, and Complete proteinase inhibitor. Cell lysates were incubated overnight with magnetic beads coated with Myc antibody (9E-10, Pierce). The immunoprecipitate was washed and run on a 4–12% Bis-Tris NuPAGE gel (Invitrogen). A slice of gel ranging between 75 and 63 kDa was excised, and tryptic peptides were prepared by *in-gel* digestion (Luís *et al*, 2016). Peptides were analyzed by nanoLC-MS using an Ekspert 425 nanoLC with cHiPLC (Eksigent, AB Sciex, Framingham, MA, USA) coupled to a TripleTOF™ 6600 mass spectrometer (AB Sciex, Framingham, MA, USA).

Spectra were searched against Swiss-Prot database (downloaded in 10/2017, 5201 entries) containing all the reviewed protein sequences available for *S. pombe*, and three human keratin sequences (P04264, P35908, P13645). The Paragon algorithm embedded in ProteinPilot 5.0 software (AB Sciex, Framingham, MA USA) was used to perform the database search using phosphorylation emphasis and gel-based ID as special factors, and biological modifications as ID focus. An independent false discovery rate (FDR) analysis was carried out using the target-decoy approach provided with Protein Pilot software, and positive identifications were achieved using a global FDR threshold below 1%.

### Two-dimensional (2D) gel electrophoresis

2D gel electrophoresis experiments were carried out as described in Noguchi *et al* (2003). 10 μg of DNA (for telomeres analysis) was digested with 60 U of *NsiI*. For analysis of the RFB region, 5 μg of

DNA was digested with 60 U of *BamHI*. DNA was run on 0.4% agarose gel for the first dimension and a 1% agarose gel for the second dimension. Gels were transferred to positively charged membranes and probed with the STE1 probe or the 1.35-kb *EcoRI-EcoRI* rDNA fragment.

**Human lentiviral infections**

HT1080 cell line was infected with either luciferase shRNA (target sequence CGCTGAGTACTTCGAAATGTC), CDS shRNA (target sequence CAAAGACCTGTTTGATCTGAT), or UTR shRNA (target sequence ACGGTAGCATTACCCAAATAA) lentiviral particles produced in 293T cells by mixing PLKO vectors with psPax2 and pVSVG vectors. Cells were infected twice and selected with puromycin at 3 μg/ml. For Fig 5B, retroviral particles were used for GFP shRNA (target sequence CACAAGCTGGAGTACAACT) or STN1 shRNA (target sequence GGGAATGGGATTTGGCATA) produced in 293T by mixing pCL-mU6 retroviral vectors with appropriate sequences with PCL-Amphotropic vector.

**siRNA transfections**

HeLa cell line was transfected with siRNA against either non-targeting (Mission siRNA Universal negative control SIC001 from SIGMA) or SSU72 (Silencer Select siRNA-1 (ID s26487) and Silencer Select siRNA-2 ID (s223816) from Ambion) using mission siRNA transfection reagent (Sigma S1452) at 10 nM following manufacturer's instructions. 3 days after transfection, cells were fixed, and IF-FISH was carried out to quantify telomere dysfunction.

**Real-Time qPCR**

To generate cDNA, RNA from HT1080 cells after 4 weeks of infection with either GFP shRNA, luciferase shRNA, and SSU72 shRNA (targeting CDS or UTR regions) was isolated using RNeasy Plus Mini Kit (Qiagen) followed by retro-transcription using NZY First-Strand cDNA Synthesis Kit. Real-time PCR was carried out using iTaq Universal SYBR Green Supermix (Bio-Rad) on the Applied Biosystems 7900 Real-Time PCR System. cDNA was amplified with specific primers for SSU72 (F-primer CGGCCGCCATTTTGTTCG R-primer CCCCGTTTGCTGAGGATGTT) or B-actin (F-primer TCCCTGGAGAA GAGCTACGA R-primer AGCACTGTGTTGGCGTACAG).

**Metaphase spreads**

Metaphases were collected by adding colcemid to the media to a 0.1 μg/ml final concentration for 4 h to overnight. Metaphases were collected using the shake off method. Cells were then incubated in hypotonic buffer (0.03M Na citrate) for 30 min and fixed with methanol: acetic acid (3:1) solution. Slides were pre-washed with 45% acetic acid spread metaphases.

**Fluorescence *in situ* hybridization (FISH)**

For FISH using telomere probes, slides were washed in PBS 3 × 5 min each with rotation and incubated with 10 mM Tris, pH 7.5 and a deionized formamide 70% telomere probe (0.5 ng/ml), blocking reagent (0.25%, 25 mM MgCl$_2$, 9 mM citric acid, 82 mM

Na2HPO4, pH 7.0) at 80°C for 3 min. Slides were then enclosed in a humidifier chamber for 3 h. After serial washes of 70% formamide, 10 mM Tris, 0.1% BSA (two times for 15 min), and PBS-Tween 0.05% (three times for 5 min), slides were dried and mounted with mounting media with 1 μg/ml DAPI. For slides with metaphase spreads, a first step of 37% pepsin digestion (0.1 g/100 ml + 88 μl HCl) was carried out before starting the FISH technique.

**Microscopy**

In fission yeast, cells were grown at 32°C in PMG with all supplements added. For each individual experiment, at least 100 cells were analyzed. For GFP and mRFP visualization, live cells were imaged using a Delta Vision Core System (Applied Precision) using a 100× 1.4 numerical aperture UplanSApo objective and a cascade2 EMCCD camera (Photometrics). Deconvolution was performed using the enhanced ratio method in softWoRx software. Co-localization experiments were performed using maximum intensity projections of deconvolved images.

For immunofluorescence-FISH analysis in human cells, cells were fixed in 2–4% formaldehyde in PBS for 10 min at RT. After SDS (0.03%) permeabilization and 15-min blocking step (1% BSA, 0.5% Triton X-100, 0.5% Tween-20), 53PB1 antibody (1:1,000; Santa Cruz biotechnology H-300) was incubated overnight. Antibody was washed three times with PBS-Tween 0.05% and a secondary antibody conjugated with Alexa 488 (1:400) was used (Badie *et al*, 2010). Before starting the FISH technique, fixation of the cells was performed using 2–4% formaldehyde for 10 min at room temperature. Images were acquired with Delta Vision Core System (Applied Precision). Deconvolution was performed using the enhanced ratio method in softWoRx software. Co-localization experiments were performed using *Z*-stacks images of deconvolved images.

# Data availability

The mass spectrometry proteomics data from this publication have been deposited to the ProteomeXchange Consortium via the PRIDE (https://www.ebi.ac.uk/pride/archive/) partner repository with the dataset identifier PXD011953.

**Expanded View** for this article is available online.

## Acknowledgements

We thank A Bianchi, AM Carr, JP Cooper, T Nakamura, P Nurse, and M Sato for sharing their strains and J Lingner and T Wang for sharing plasmid constructs. We are grateful to K Tomita, L Jansen, R Carlos, and A Sridhar for reading the manuscript and to MGF laboratory for critical comments and discussions. This work was supported by the Portuguese Fundação Ciência e tecnologia (FCT) project number PTDC/BEX-BCM/5179/14. JME is supported by PTDC/BEX-BCM/5179/14 and PIEF-GA-2013-624759, and ESMC is supported by SFRH/BD/113754/2015. SM is supported by the Ligue Nationale Contre le Cancer (LNCC, Equipe Labellisée Vincent Géli). SC is supported by Projet Fondation ARC and by the Agence Nationale de la Recherche (ANR-16-CE12 TeloMito). IML acknowledges FCT for PhD fellowship funding (PD/BD/113982/2015) under the MolBioS PhD-Program (PD/00133/2012). IAA acknowledges IF/00764/2014 Research unit GREEN-IT "Bioresources for Sustainability" (UID/Multi/04551/

2013). Mass spectrometry analyses were performed at UniMS (ITQB/iBET). MGF was supported by the HHMI International Early Career Scientist program.

## Author contributions

MG-F and JME conceived the study and designed the experiments. JME and ESMC performed the majority of the experiments. MG and CCR performed the fission yeast genetic screen. SC and SM performed the 2D gel experiments. IML and IAA performed the mass spectrometry analysis. MGF and JME wrote the manuscript. MGF supervised the research.

## Conflict of interest

The authors declare that they have no conflict of interest.

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
