## [Review Process File · The EMBO Journal]

Ssu72 phosphatase is a conserved telomere replication terminator

Jose Miguel Escandell, Edison S. M. Carvalho, Maria Gallo-Fernandez, Clara C. Reis, Samah Matmati, Inês Matias Luís, Isabel A. Abreu, Stéphane Coulon and Miguel Godinho Ferreira.

Review timeline:

Submission date:	13 th August 2018
Editorial Decision:	21 st September 2018
Revision received:	18 th December 2018
Editorial Decision:	9 th January 2019
Revision received:	21 st January 2019
Accepted:	28 th January 2019

Editor: Hartmut Vodermaier

Transaction Report:

1st Editorial Decision

21st September 2018

Thank you for submitting your manuscript on SSU72 roles in telomere replication for our editorial consideration. With some delay linked to the summer vacation season, we have now obtained a complete set of reports from three expert referees, copied below for your information. I am pleased to inform you that all referees acknowledge the interest and potential importance of your findings, as well as the overall technical quality of the study. They nevertheless bring up a number of important points that would need to be addressed prior to eventual publication. The most significant concern in this respect pertains to the data on human SSU72 and conservation of its telomere regulation role, which both referees 2 and 3 feel would require further experimental backup to fully support such conclusions and to clarify whether SSU72 acts in the same or in somewhat different ways in the two systems. Another major issue raised by referee 2 relates to the conclusiveness of the data supporting CDK involvement in Stn1 phosphorylation.

Should you be able to satisfactorily address these key concerns, as well as the various other more specific experimental and textual issues noted by all three referees, we should be happy to consider a revised manuscript further for publication in The EMBO Journal.

REFeree REPORTS

Referee #1:

The CST complex plays crucial roles in regulating telomerase and mediating fill-in synthesis of the telomeric lagging strand. In this paper, the authors gain important insights into the regulation of the CST complex. It is shown in fission yeast that Ssu72 phosphatase regulates Stn1 recruitment and telomere replication through dephosphorylation of Stn1 at serine 74.

First, the authors show that *ssu72Δ* cells have much longer telomeres (~1kb) than wild type cells and that Ssu72 is recruited to telomeres in late S-phase. Telomere elongation is demonstrated to be

telomerase-dependent and native gel analysis indicated increased single strandedness of the G-rich strand, suggesting defects in lagging strand synthesis. Consistently, a mutant carrying a hypomorphic *stn1-75* allele showed the same telomere elongation phenotype as *ssu72Δ* cells and *stn1-75 ssu72Δ* double mutant cells. Further, it is demonstrated that Stn1 recruitment to telomeres in late S/G2 is dependent on Ssu72. 2D-gel analyses are used to assess telomere replication. It is claimed that replicative Y structures were not observed at telomeres in *ssu27Δ* cells though the data is not convincing to this referee (see specific comments). Next it is shown that a phosphomimetic *stn1-S74D* mutant has the same telomere length as *ssu72Δ* cells and *stn1-S74D ssu72Δ* double mutant cells. This result provides excellent evidence that Ssu72 regulates telomere length through dephosphorylation of Stn1 at Ser 74. It is corroborated in subsequent experiments that this effect is mediated through Stn1 recruitment, which in turn promotes DNA pol alpha-primase. Epistasis experiments with a *pol1*-mutants show that Ssu72 cooperates with DNA pol alpha-primase. Furthermore, a physical interaction, which was detected between Pol1 and Stn1 by co-immunoprecipitation was dependent on Ssu72. Finally, the findings were extended to human cells. It is shown that depletion of SSU72 in HT1080 cells leads to telomerase-dependent telomere elongation and increased telomere fragility suggesting telomere replication defects as seen in fission yeast.

Overall, the experimental data is very extensive and in most parts convincing. The gained knowledge is very important.

Specific comments:

Figure 3B, page 10. It is concluded from the 2D gel analysis that telomere replication but not rDNA replication is defective in *ssu27Δ* cells. However, the replication fork structures seem also much weaker for the rDNA control in *ssu27Δ* cells. Thus, it seems that loading of DNA in wild type and mutant strains is unequal or the control is not well chosen. In either case, the experiment is not conclusive for this referee and must be improved or eliminated.

Minor comments:

Reference annotation is inconsistent: Mostly with numbers within the text but on page 3 and 4 with author name and year (e.g. on page 4: Chang et al., 2013).

Referee #2:

The action of telomerase at telomeres is regulated in coordination with replication of the bulk of the telomere. In particular, the CST complex is critical for the last steps of telomere replication and is understood to inhibit telomerase activity at least in part through the promotion of C-strand synthesis via stimulation of PolAlpha/Primase activity. This study identifies the phosphatase Ssu72 as a regulator of telomerase action at fission yeast telomeres. Ssu72 is shown to be required to control the phosphorylation status of a specific residue of Stn1. Stn1 phosphorylation in turn is indicated to control both Stn1 association to telomeres and its interaction with Pol1. Although uncertainty remains with regard to the kinase involved in this pathway, taken together the results assemble a compelling picture of how control of phosphorylation status of Stn1 by Ssu72 is required to limit telomerase action in the later stages of telomere replication. This level of control seems to be conserved in human cells, as the experiments presented suggest. Specifically, down-regulation of human SSU72 leads to telomere elongation (in a telomerase-dependent manner) and to loss of STN1 from telomeres, and also mimics some of the STN1 phenotypes in human cells.

I think this is a well-executed and important study, that introduces a relevant new player in telomere replication. I have two main reservations. First, it would have been nice to see a complementation of the human SSU72 phenotype with wild-type and mutant SSU72 constructs. Second, I am rather dubious about the evidence supporting direct CDK involvement in the phosphorylation of Stn1 (see below).

Minor points:

I would make a clear distinction, for the sake of clarity, between 'telomere replication' (by the

replication fork) and 'telomere extension' - or similar - (by telomerase).

130 Western blot shift analysis, we observed that Ccq1 was phosphorylated in *ssu72Δ*
Change this to 'hyper-phosphorylated', as I think the authors here imply that the phosphorylation
is increased in the mutant compared to wild-type (previous work had demonstrate that
phosphorylation does occur in wt cells).

153 similar telomere lengths. Further, we consistently detected Rad1 IRPA-GFP
154 localization at telomeres, as measured by live imaging in *ssu72Δ* mutant cells
Poorly phrased.

239 telomeres (Figure 3E). Taken together, our data suggest that fission yeast Stn1 is
240 phosphorylated at Serine-74 to enable its efficient recruitment to telomeres and,
241 consequently, efficient DNA replication and telomerase regulation.
This statement seems to suggest that phosphorylation promotes Stn1 telomere association, the
opposite of what the data show. (The same error applies to line 243.)

In figure S3 it should be 'S. cryophilus'. The legend to this figure is incomplete, and the writing
could benefit from a second look.

I find the *cdc2* experiments rather unconvincing. To begin with, the densitometry data should have a
label on their y axis (it is not clear whether these data belong to panel B, or to a third panel). The
effects are small and presumably derived from a single experiment. Are *ssu72* mutants altered in
their cell cycle behaviour? Do they divide more slowly? Are the cells enlarged, possibly indicating
DDR activation? (also, in the legend, I suppose it should be 'for 16 hours' rather than 'by 16 hours?')
Obviously, *cdc2* has many targets, and the small effect seen here might be indirect. I would suggest
toning down the conclusion in the main text, or performing additional experiments, for example
using *cdc2-as* alleles and looking at a more direct readout than telomere length, such as Stn1
telomere association.

Given that a quantification, and repeats, are missing from the telomere blot in figure 4A, it is hard to
tell whether *ssu72D* is epistatic to *pol1-13*. In any case, *pol1-13* is a hypomorph and therefore
epistasis need to be taken with a pinch of salt. Similar considerations apply to the data in figure 5A.
Overall, it appears that the double mutant has slightly longer telomeres compared to either single
mutant, which could suggest that *ssu72* partly functions independently of *polalpha/primase* or that
the *ssu72* exacerbates the phenotype of the hypomorphs.

The *hsRNA* experiment in human cells, which leads the authors to conclude that *SSH72* acts with
STN1 in human cells on the basis of results from the literature and obtained in other cells lines,
should have been conducted with down-regulation of *STN1* as well. Epistasis analysis of the double-
hit cells would allow firmer conclusions.

Referee #3:

The focus of these studies was to identify regulators of telomere homeostasis in fission yeast. In
their genome-wide deletion screen, they identified *SSU72*, a conserved phosphatase known to
function in the regulation of RNA polymerase II and sister chromosome cohesion. Their study goes
on to show that *SSU72* regulates telomere length in a telomerase-dependent manner through
dephosphorylation of Ccq1, which is required for telomerase recruitment, and Stn1, which inhibits
continued telomerase extension. They then show that the increase in telomere length in the
ssu72delta cells is caused by an inability of Stn1 to localize to the telomere, resulting telomere
elongation and ssDNA. This is shown through a series of experiments which demonstrate that Stn1
is localized to the telomere in a *SSU72* dependent manner, that *SSU72* activity is required for Stn1
interaction with DNA polymerase alpha-primase and that *SSU72* acts at a conserved residue (S74)
in Stn1. Creation of phosphomimetic Stn1 S74D phenocopies the increased telomere length in
ssu72delta cells. They then attempt to show conservation of *SSU72* function in humans. Overall, the
studies are well constructed, the manuscript well written and the findings could significantly
increase our understanding of telomere length regulation from yeast to humans. However, the data
to show conservation between the fission yeast and human cellular studies are incomplete. In my

opinion, further studies to clarify the role of the SSU72 in human cells are needed to increase the impact and significance of the findings for publication in EMBO J. Below are specific comments that I feel should be addressed prior to publication. If properly addressed, these studies would be of significant impact to our understanding of telomere biology and identify a novel player in the regulation of telomere homeostasis.

1. My major comments are in relation to the overall conclusions drawn from the study. In the first part of the manuscript, the focus is on changes in G-overhang length and the role of Stn1-Ten1. The focus is then shifted to telomere duplex replication defects in human cells, in the form of telomere fragility (MTS), without assessing G-overhang length. In humans, passage of the replication fork and telomere extension has been shown to occur prior to C-strand fill-in (Zhao, et al. Cell, 2009). From the current results, I infer that SSU72 regulates overhang length, or lagging strand synthesis, in fission yeast but only telomere duplex replication in humans. It may likely be involved in G-overhang regulation (i.e. telomerase inhibition and/or C-strand fill-in) in humans but no experiments to assess G-overhang length were performed. Thus, it is unclear how conserved its function is. These studies have also not directly linked SSU72 activity to STN1 function in humans, which could be determined by mutation of the conserved residue (T81) in STN1. Further experiments are needed to address the effects of SSU72 on G-overhang regulation in humans. This could include:

- a. G-overhang analysis with and without ExoI treatment to demonstrate whether any increases arise from the G-overhang or internal ssDNA regions.
- b. Interaction between STN1 and DNA polymerase alpha should also be assessed in the shSSU72 cells, similar to what was done in yeast.
- c. Add back of a STN1 T81 phosphomimetic mutant to STN1 depleted cells to show that this residue is indeed conserved and mimics the knockdown of SSU72.

2. Figure 2: It is unclear why the background Rad11-GFP signal is so high compared to the control. The intensity/exposure is expected to be similar across the samples and the high background could affect the number of foci detected.

3. The title for results section (line 145) for Figure 2 states "Ssu72 phosphatase function is independent of Rif1 and Taz1/Rap1/Poz1". Results for Rif1 are shown but there is no data for Taz1/Rap1/Poz1. It is unclear why this is included in the title.

4. Figure 2A: Since later results with the shSSU72 demonstrate duplex replication defects, a ExoI digested control is important to show that the signal is G-overhang and not internal ssDNA.

5. Figure 2C: The authors conclude that telomere length regulation is independent of Rif1. However, the telomere length increase is not additive but appears to be synergistic, suggesting that Rif1 and SSU72 converge to regulate telomere length. Discussion of this should be included in the manuscript.

6. Figure 3C: The data would be strengthened by including a stn1-D74A mutant, which presumably would shorten telomere length by inhibiting telomerase early in the process.

7. Figure 4D: The authors state on line 305-306 that "...overexpression of pol1+ in ssu72delta mutants is sufficient to rescue telomere defects." However, the rescue, while significant, is only partial, suggest that other mechanisms may be at play (e.g. telomerase inhibition).

8. Figure 5A: The increase in telomere length is modest at best. Replicates and a graph of the change in telomere length should be shown to demonstrate a significant increase in telomere length. Assessment of the knockdown levels of SSU72 are not shown and should be included.

9. Figure 5B and S6A: The representative telomere FISH images shown suggest that the FISH quality may not be adequate to properly count MTS. It appears that many of the chromosomes have signal-free ends (SFEs) and there is significant background in the images that would make scoring difficult. Was the scoring of the MTS blinded? Does each dot in the graph represent a metaphase spread? Are there changes in SFEs or fusions with SSU72 knockdown?

10. Line 353: The authors state that "...STN1 downregulation in human also results in increased MTS." However, many other proteins also lead to MTS/telomere fragility so it is possible that SSU72

affects other replication or telomere proteins. As stated above, mutation of STN1 at T81, or other methods, to show a direct link between SSU72 phosphatase activity on STN1 are needed.

11. Line 406: "... ssu72delta cells have longer overhangs, extensive phosphorylation of Ccq1 and higher levels of telomerase at telomeres." It is unclear what piece of data shows higher levels of telomerase at telomeres.

Minor:

1. Figure 1D: *trt1delta* is mentioned as a control but not shown in the figure.
2. Figure 1F: No loading control is included.
3. Line 168: The superscript CTC1 with Cdc13 does not seem appropriate, as functional conservation between Cdc13 and CTC1 is not yet established.
4. Figure 3A: Untagged control was only included for one timepoint and not across the timecourse.
5. Line 331-332: The sentence needs a qualifier (e.g. We wondered whether the observed telomere elongation...)

Reply to referees' comments:**Referee #1:**

(...)

Overall, the experimental data is very extensive and in most parts convincing. The gained knowledge is very important.

Specific comments:

Figure 3B, page 10. It is concluded from the 2D gel analysis that telomere replication but not rDNA replication is defective in *ssu27Δ* cells. However, the replication fork structures seem also much weaker for the rDNA control in *ssu27Δ* cells. Thus, it seems that loading of DNA in wild type and mutant strains is unequal or the control is not well chosen. In either case, the experiment is not conclusive for this referee and must be improved or eliminated.

We thank the Referee #1 for this fair comment. A new experiment is now included in Figure 3B. Strikingly, *ssu72Δ* cells have lower replication forks intermediates than WT cells at the rDNA. This data is consistent with previous results published by the Ishikawa laboratory (Takikawa et al., 2017 NAR). In this manuscript, the authors show that *Stn1* inactivation resulted in lower replication fork intermediates at the rDNA locus as compared to WT. Altogether, our experiments suggest that *Ssu72* not only controls *Stn1* at telomeres, but also at the rDNA locus and, possibly, throughout the genome. These results are now included in Results section.

Minor comments:

Reference annotation is inconsistent: Mostly with numbers within the text but on page 3 and 4 with author name and year (e.g. on page 4: Chang et al., 2013).

We thank the Referee #1 for detecting this typo. We have duly corrected it.

Referee #2:

(...)

I think this is a well-executed and important study, that introduces a relevant new player in telomere replication. I have two main reservations. First, it would have been nice to see a complementation of the human *SSU72* phenotype with wild-type and mutant *SSU72* constructs. Second, I am rather dubious about the evidence supporting direct CDK involvement in the phosphorylation of *Stn1* (see below).

We thank Referee #2's comments and suggestions. We concentrated on Referee #2's critical second point and provided further evidence for *Cdc2* involvement in *Stn1*

phosphorylation. As recommended, we used the *cdc2-as* allele to inactivate Cdk1 activity and measured Stn1 recruitment to telomeres in the absence of *ssu72Δ* with positive results. We thank Referee #2 for this insightful experiment, and we have included this new data in the manuscript in Expanded View Figure 3 and below in Explanatory Figure 2 (see below for complete description).

Minor points:

I would make a clear distinction, for the sake of clarity, between 'telomere replication' (by the replication fork) and 'telomere extension' - or similar - (by telomerase).

We thank Referee #2 for the suggestion and we clarified these statements throughout the manuscript.

130 Western blot shift analysis, we observed that Ccq1 was phosphorylated in *ssu72Δ*
Change this to 'hyper-phosphorylated', as I think the authors here imply that that the phosphorylation is increased in the mutant compared to wild-type (previous work had demonstrated that phosphorylation does occur in wt cells).

We changed the text according to Referee #2's remark.

153 similar telomere lengths. Further, we consistently detected Rad11RPA-GFP
154 localization at telomeres, as measured by live imaging in *ssu72Δ* mutant cells
Poorly phrased.

We changed the text according to Referee #2's comments.

telomeres (Figure 3E). Taken together, our data suggest that fission yeast Stn1 is phosphorylated at Serine-74 to enable its efficient recruitment to telomeres and, consequently, efficient DNA replication and telomerase regulation. This statement seems to suggest that phosphorylation promotes Stn1 telomere association, the opposite of what the data show. (The same error applies to line 243.)

We thank the Referee #2 for detecting this typo. It has been corrected in the current manuscript.

In Figure S3 it should be '*S. cryophilus*'. The legend to this Figure is incomplete, and the writing could benefit from a second look.

The new version is corrected and the text is rewritten following Referee #2's comment.

I find the *cdc2* experiments rather unconvincing. To begin with, the densitometry data should have a label on their y axis (it is not clear whether these data belong to panel B, or to a third panel). The effects are small and presumably derived from a single experiment.

We thank Referee #2's comments. We have replaced densitometry data for proper Telomere Restriction Fragment analysis (TRF) in Expanded View Figure 3A. In addition, we would like to point out that this experiment was repeated several times using similar conditions. In **Explanatory Figure 1**, we included further examples of temperature titration experiments and corresponding TRF analysis quantification.

Explanatory Figure 1

Explanatory Figure 1: *cdc2* inactivation rescues telomere defect in *ssu72Δ* cells.

A) Cells were grown at 25°C and then shifted to different temperatures for 16 hours to partially inactivate *cdc2*-M68. DNA was isolated and telomere length measured with *Apa*I digested DNA. Independent experiments and quantification of Telomere Restriction Fragment analysis (TRFs) was performed on individual samples.

B) Full gel from A). Relevant samples are represented in red outlines

Are *ssu72* mutants altered in their cell cycle behavior? Do they divide more slowly? Are the cells enlarged, possibly indicating DDR activation?

Referee #2 is right. Indeed, a minority of *ssu72Δ* mutants are longer than WT and exhibit Rad11-GFP foci both telomeric and non telomeric. However, using PI FACS analysis, we do not observe major differences in cell cycle profile (Expanded Figure 3C), suggesting that *ssu72Δ* cell cycle is not grossly affected.

(also, in the legend, I suppose it should be 'for 16 hours' rather than 'by 16 hours'?)

We thank the Referee #2 for detecting this typo. We changed the text according to Referee #2's remark.

Obviously, *cdc2* has many targets, and the small effect seen here might be indirect. I would suggest toning down the conclusion in the main text, or performing additional experiments, for example using *cdc2-as* alleles and looking at a more direct readout than telomere length, such as Stn1 telomere association.

We were encouraged by Referee #2's suggestion and decided to use *ssu72Δ cdc2-as* double mutants to perform the recommended experiment. Strikingly, whereas in *ssu72Δ* mutants we do not observe Stn1 recruitment to telomeres, upon *cdc2-as* inactivation by adding ATP analog 1NM-PP1, we greatly rescue Stn1 recruitment to telomeres (Expanded View Figure 3B). Therefore, this result strongly suggests that Ssu72 dependent Stn1 recruitment to telomeres is counteracted by Cdc2 kinase activity. This data is now included in Expanded View Figure 3.

Nevertheless, we duly acknowledge that our result may still constitute an indirect effect of Cdc2 on Stn1. However, we do not believe it to be a global *cdc2* cell cycle defect. During the development of this manuscript, we tested several kinases related to previously described SSU72 functions to uncover which would compensate for *ssu72Δ* telomere length defects. We used mutants for both RNA Pol1 C-Terminal Domain (CTD) kinase Msc6 and cell cycle kinases (Hsk1 and Plo1). We tested if these mutations would result in suppression of telomere length defect of *ssu72Δ* mutants. In Explanatory Figure 2, we show that none of these kinases were able to compensate for telomere elongation of *ssu72Δ* mutants, as observed in *cdc2-M68 ssu72Δ*.

Explanatory Figure 2

Explanatory Figure 2: Inactivation of several kinases does not rescue telomere length defects of *ssu72Δ* mutants.

A) *wt*, *plo1-41*, *ssu72Δ* and *plo1-41 ssu72Δ* double mutants were grown at 25°C and then shifted to 32°C for 16 hours to partially inactivate Plo1 kinase. DNA was isolated and telomere length measured with *Apal* digested DNA.

B) Full gel from A). Relevant samples are outlined in red

C) *wt*, *hsk1-1312*, *ssu72Δ* and *hsk1-1312 ssu72Δ* double mutants were grown at 25°C and then shifted to 32°C for 16 hours to partially inactivate Hsk1 kinase. DNA was isolated and telomere length was measured with *Apal* digested DNA.

D) Full gel from D). Relevant samples are outlined in red

E) *wt*, *mcs6-13*, *ssu72Δ* and *mcs6-13 ssu72Δ* double mutant cells were grown at 25°C and then shifted to 32°C for 16 hours to partially inactivate *Mcs6*. DNA was isolated and telomere length measured with *Apal* digested DNA.

Given that a quantification, and repeats, are missing from the telomere blot in Figure 4A, it is hard to tell whether *ssu72D* is epistatic to *pol1-13*. In any case, *pol1-13* is a hypomorph and therefore epistasis needs to be taken with a pinch of salt. Similar considerations apply to the

data in Figure 5A. Overall, it appears that the double mutant has slightly longer telomeres compared to either single mutant, which could suggest that *ssu72* partly functions independently of *pol1-13*/primase or that the *ssu72* exacerbates the phenotype of the hypomorphs.

In **Explanatory Figure 3**, we included three previous repeats for evaluation. We carefully analyzed several gels by TRF analysis from independent DNA isolations. Medians from different experiments were averaged and statistical analysis was carried out. Whereas we could detect difference between *ssu72Δ/wt* mutants, we were unable to detect statistically significant differences between *pol1-13* and *pol1-13 ssu72Δ* double mutants.

Explanatory Figure 3

Explanatory Figure 3: *ssu72Δ* is epistatic with *pol1-13* for telomere length regulation (repeats)

A) *ssu72*⁺ and DNA polymerase α regulate telomere length in the same genetic pathway. Epistasis analysis of *ssu72Δ* and *pol1-13* performed by Southern blotting of *Apal* digested DNA using a telomeric probe. TRF analysis was carried out and medians calculated for 3 different experiments. B) Quantification of Medians: n=3; **p* ≤ 0.05 based on a two-tailed Student's t-test to control sample. Error bars represent standard error of the mean (SEM).

In addition, we performed *EcoRI* digestions using genomic DNA derived from further *pol1-13* and *pol1-13 ssu72Δ* mutants and showed to have very similar telomere length (**Explanatory Figure 4**). Altogether, to the best of our analysis, *pol1-13* and *ssu72Δ* mutants appear to be epistatic for telomere length regulation.

Explanatory Figure 4

Explanatory Figure 4: *ssu72Δ* is epistatic with *pol1-13* for telomere length regulation (EcoRI genomic DNA digest)

A) *ssu72⁺* and DNA polymerase α regulate telomere length in the same genetic pathway. Epistasis analysis of *ssu72Δ* and *pol1-13* performed by Southern blotting of *EcoRI* digested DNA using a telomeric probe.

The hsRNA experiment in human cells, which leads the authors to conclude that SSH72 acts with STN1 in human cells on the basis of results from the literature and obtained in other cells lines, should have been conducted with down-regulation of STN1 as well. Epistasis analysis of the double-hit cells would allow firmer conclusions.

We thank both Referee #2 and Referee #3 for this suggestion. We have performed epistasis analysis for human SSU72 and STN1 and we have now included it in the manuscript as Figure 5B. Our results show that, while SSU72 shRNA treated cells and

STN1 shRNA treated cells have longer telomeres than WT, double SSU72/STN1 shRNA treated cells have same telomere length as STN1 shRNA treated cells. This result reinforces our model in which human STN1 and SSU72 control telomere length in the same genetic pathway.

Referee #3:

(...)

Overall, the studies are well constructed, the manuscript well written and the findings could significantly increase our understanding of telomere length regulation from yeast to humans. However, the data to show conservation between the fission yeast and human cellular studies are incomplete. In my opinion, further studies to clarify the role of the SSU72 in human cells are needed to increase the impact and significance of the findings for publication in EMBO J. Below are specific comments that I feel should be addressed prior to publication. If properly addressed, these studies would be of significant impact to our understanding of telomere biology and identify a novel player in the regulation of telomere homeostasis.

1. My major comments are in relation to the overall conclusions drawn from the study. In the first part of the manuscript, the focus is on changes in G-overhang length and the role of Stn1-Ten1. The focus is then shifted to telomere duplex replication defects in human cells, in the form of telomere fragility (MTS), without assessing G-overhang length. In humans, passage of the replication fork and telomere extension has been shown to occur prior to C-strand fill-in (Zhao, et al. Cell, 2009). From the current results, I infer that SSU72 regulates overhang length, or lagging strand synthesis, in fission yeast but only telomere duplex replication in humans. It may likely be involved in G-overhang regulation (i.e. telomerase inhibition and/or C-strand fill-in) in humans but no experiments to assess G-overhang length were performed. Thus, it is unclear how conserved its function is. These studies have also not directly linked SSU72 activity to STN1 function in humans, which could be determined by mutation of the conserved residue (T81) in STN1. Further experiments are needed to address the effects of SSU72 on G-overhang regulation in humans.

We thank Referee #3's comments and suggestions. We have to agree with Referee #3 that we have not shown that the conserved function of SSU72 from fission yeast to humans is on regulation telomere lagging strand synthesis. However, we have strong evidence that SSU72 regulates STN1 recruitment to telomeres in both organisms and, consequently, suggesting that it may regulate STN1 dependent functions at telomeres.

Four main results encourage us to suggest this:

1. We observe telomere elongation upon reduction of SSU72 (Figures 1A in fission yeast and 5A in humans)
2. Recently, two published manuscripts (Takikawa *et al*, 2017 NAR; Matmati *et al*, 2018 Sci. Adv) in fission yeast showed that, in addition to telomerase regulation, Stn1 is required for telomere and subtelomere replication. Our own 2D gel analysis shows that Ssu72 controls DNA replication both at telomeres and rDNA loci, suggestive of a broader genomic function. In agreement, SSU72 depletion induces telomere fragility (presumably stemming from telomere replication defects) (Figures 3B in fission yeast and 5B in humans)
3. SSU72 depletion induces DNA Damage Response at telomeres. (Figures 2B in fission yeast and 6A in humans)
4. STN1 recruitment to telomeres is defective upon SSU72 depletion (Figure 2A in fission yeast and 6B in humans)

In addition to these results, we are excited to present in this version of our manuscript new evidence suggesting that SSU72 and STN1 control telomere length in the same genetic pathway in humans. Please, see our answer to Referee #2's final remark.

This could include:

- a. G-overhang analysis with and without Exol treatment to demonstrate whether any increases arise from the G-overhang or internal ssDNA regions.

We thank Referee #3 for this suggestion. Upon Referee #3's request, we carried out overhang quantification of SSU72 downregulated cells using Exol treatment. SSU72 downregulated cells have significant longer overhangs compared to controls. We have now included this data in new version of the manuscript (Appendix S5).

- b. Interaction between STN1 and DNA polymerase alpha should also be assessed in the shSSU72 cells, similar to what was done in yeast.

According to Referee #3's suggestion, we initially performed this experiment in HT1080 cells, the cell line used throughout our study. Unfortunately, we were unable to immunoprecipitate STN1 using POLA2 (p180) antibodies (Dioti *et al.*, 2016 Mol Cancer Res) in control experiments, thus making it impossible to test this interaction in SSU72 depleted cells (Explanatory Figure 5A). In a second attempt, we tried to perform this experiment using Hek293T transfected with STN1-FLAG. We tested if we could immunoprecipitate POLA2 (p180) with STN1-FLAG and we were unable to detect this interaction in otherwise unperturbed cells (Explanatory Figure 5B). In conclusion, using two independent systems, we were unable to test this interesting experiment suggested by Referee #3.

Explanatory Figure 5

Explanatory Figure 5: Polymerase alpha p180 catalytic subunit failed to co-purify with STN1

A) HT1080 cell line extracts were prepared and p180 subunit was immunoprecipitated using p180 specific serum. Endogenous proteins were detected by western blot.

B) HEK293T cells were transfected with STN1-FLAG construct by calcium precipitation method. Cell extracts were incubated with pre-immune serum or serum raised against p180 subunit and processed for immunoprecipitation. Western blot analysis failed to detect STN1 in either experiment. Long and short refer to different exposures.

- a. Add back of a STN1 T81 phosphomimetic mutant to STN1 depleted cells to show that this residue is indeed conserved and mimics the knockdown of SSU72.

We thank the Referee #3 for this suggestion. We set up all reagents to carry out the intended experiment in HT1080 cells. However, in our preliminary experiments, we were unable to complement telomere elongation resulting from STN1 depletion by expression of WT STN1-FLAG (Explanatory Figure 6A and B). STN1 expression was subsequently validated by western blotting using STN1 monoclonal antibodies. We noticed that our experimental setup grossly overexpressed STN1 (Explanatory Figure 6C), rendering these results difficult to interpret in a conclusive manner. We envisage gene-editing tools as the best approach to test the role of this residue in STN1 regulation. However, these techniques are not possible to carry out within the time frame of this revision.

Explanatory Figure 6

Explanatory Figure 6: STN1 overexpression does not complement STN1 shRNA depleted cells

A) HT1080 cells were infected with retroviral particles for STN1-Flag and shRNAs against STN1 (UTR region) and control GFP shRNA for 4 weeks. Genomic DNA was collected and analysed by Southern blot for telomere length.

B) Quantification of Telomere Restriction Fragment analysis (TRFs). Medians are calculated and presented above.

C) STN1 western blot of cells expressing either pbabe-GFP or STN1-FLAG constructs infected with STN1 shRNA or GFP shRNA. Tubulin was used as loading control. Long and short refer to different exposures.

2. Figure 2: It is unclear why the background Rad11-GFP signal is so high compared to the control. The intensity/exposure is expected to be similar across the samples and the high background could affect the number of foci detected.

Depending on expression levels, live analysis of endogenous fission yeast proteins tagged under their own promoter typically results in higher background. We have now

substituted the previous images with new ones and added insets for clarity. In addition, insets were added for better visualization. We would like to point out that quantification was performed using similar background levels for all samples.

3. The title for results section (line 145) for Figure 2 states "Ssu72 phosphatase function is independent of Rif1 and Taz1/Rap1/Poz1". Results for Rif1 are shown but there is no data for Taz1/Rap1/Poz1. It is unclear why this is included in the title.

We thank Referee #3 for pointing this oversight. We have corrected it in the current manuscript version.

4. Figure 2A: Since later results with the shSSU72 demonstrate duplex replication defects, a ExoI digested control is important to show that the signal is G-overhang and not internal ssDNA.

According to Referee #3's suggestion, we have included an ExoI treatment control for fission yeast *ssu72Δ* overhang assays. We have added this control to the results section and it is shown in Appendix Figure S3.

5. Figure 2C: The authors conclude that telomere length regulation is independent of Rif1. However, the telomere length increase is not additive but appears to be synergistic, suggesting that Rif1 and SSU72 converge to regulate telomere length. Discussion of this should be included in the manuscript.

We thank Referee #3 for this fair assessment. The evident synergistic effect between *ssu72Δ* and *rif1Δ* mutants is now included in the results section.

6. Figure 3C: The data would be strengthened by including a *stn1-D74A* mutant, which presumably would shorten telomere length by inhibiting telomerase early in the process.

Similar to Referee #3's suggestion, we also hypothesized that *stn1-S74A* mutant could have shorter telomeres. To test this, we constructed a *stn1-S74A* mutant strain and measured its telomere length. We observed that, in contrast to our expectation, *stn1-S74A* mutants have similar long telomeres as *stn1-S74D* mutants (Explanatory Figure 7). This type of phenomenon has been previously observed by others, such that both alanine and aspartic acid mutants give rise to similar terminal phenotypes (e.g. Yamazaki *et al*, 2012 Genes and Dev). One potential explanation may involve this regulatory step to constitute a continuous switch that, once locked in either position - on or off - would block the entire process, thus rendering a similar terminal phenotype. We decided to exclude this data until further experiments are carried out to test this hypothesis.

Explanatory Figure 7

Explanatory Figure 7: Both *stn1-S74D* and *stn1-S74A* mutations elongate telomeres

Serine 74 substitution to a phosphomimetic aspartate amino acid (*stn1-S74D*) or phospho mutant alanine (*stn1-S74A*) elongate telomeres. WT and *stn1-S74D* or *stn1-S74A* mutants were analysed by Southern blotting using *EcoRI* digested genomic DNA and a telomeric probe.

7. Figure 4D: The authors state on line 305-306 that "...overexpression of *pol1+* in *ssu72delta* mutants is sufficient to rescue telomere defects." However, the rescue, while significant, is only partial, suggest that other mechanisms may be at play (e.g. telomerase inhibition).

We agree with Referee #3's comment and acknowledge the insight. The sentence has now changed to "(...) overexpression of *pol1+* in *ssu72Δ* mutants partially rescued telomere defects."

8. Figure 5A: The increase in telomere length is modest at best. Replicates and a graph of

the change in telomere length should be shown to demonstrate a significant increase in telomere length. Assessment of the knockdown levels of SSU72 are not shown and should be included.

We acknowledge Referee #3's comment and we would like to point out that the knockdown efficiency was, indeed, indicated in the results section. However, following Referee #3's suggestion, we have now included them in Figure 5A. In addition, replicas of the experiment are included below in **Explanatory Figures 8 and 9**.

Explanatory Figure 8

Replica 1 time response

Explanatory Figure 8: Down-regulation of human SSU72 results in a time dependent telomere elongation (replica 1).

A) HT1080 cells were infected with lentiviral particles carrying two independent shRNAs against SSU72 (CDS and UTR regions) and control Luciferase (Luc) shRNA for 2, 4 and 6 weeks and DNA was collected to test by Southern blot telomere length.

B) Quantification of Telomere restriction fragment analysis (TRFs) at the 6 weeks' time point.

Explanatory Figure 9

Explanatory Figure 9: Down-regulation of human SSU72 results in a time dependent telomere elongation (replica 2).

A) HT1080 cells were infected with lentiviral particles carrying two independent shRNAs against SSU72 (CDS and UTR regions) and control Luciferase (Luc) shRNA for 2, 4 and 6 weeks and DNA was collected to test by Southern blot telomere length.

B) Quantification of Telomere restriction fragment analysis (TRFs) at the 6 weeks' time point.

In addition, we were able to recapitulate the same phenotype in a different cell line, RPE cells (Explanatory Figure 10). We consider this data preliminary and we would not include it in the final manuscript.

Explanatory Figure 10

Explanatory Figure 10 Down-regulation of human SSU72 results in telomere elongation in RPE cells

A) RPE cells infected with lentiviral particles carrying two independent shRNAs against SSU72 (CDS and UTR regions) and control Luciferase (Luc) shRNA for 4 weeks.

B) Quantification of Telomere Restriction Fragment analysis of A (TRFs).

9. Figure 5B and S6A: The representative telomere FISH images shown suggest that the FISH quality may not be adequate to properly count MTS. It appears that many of the chromosomes have signal-free ends (SFEs) and there is significant background in the images that would make scoring difficult. Was the scoring of the MTS blinded? Does each

dot in the graph represent a metaphase spread? Are there changes in SFEs or fusions with SSU72 knockdown?

Referee #3 is right in stating that our data acquisition was not taken blindly. We have now corrected, and a new experiment was carried out and analyzed blindly obtaining similar results. This experiment was added to our data set in Figure 5C. Every dot represents a metaphase quantified from 4 different experiments. In addition, we didn't observe SFE or fusions in SSU72 shRNA treated cells.

10. Line 353: The authors state that "...STN1 downregulation in human also results in increased MTS." However, many other proteins also lead to MTS/telomere fragility so it possible that SSU72 affects other replication or telomere proteins. As stated above, mutation of STN1 at T81, or other methods, to show a direct link between SSU72 phosphatase activity on STN1 are needed.

We thank the Referee #3 for this suggestion. As mentioned above, due to inability of complementing STN1 downregulation using a WT STN1 construct that was grossly overexpressed, we were discouraged to mutate STN1 to STN1-T81D. We believe the best approach for these experiments will require gene editing tools to manipulate the endogenous locus and test this interesting hypothesis. However, it was not possible to performed it in the time frame of this revision.

11. Line 406:"... ssu72delta cells have longer overhangs, extensive phosphorylation of Ccq1 and higher levels of telomerase at telomeres." It is unclear what piece of data shows higher levels of telomerase at telomeres.

We show in Figure 1E that telomerase (Trt1-myc) is enriched at *ssu72Δ* telomeres as observed by Chromatin Immunoprecipitation experiments.

Minor:

1. Figure 1D: *trt1delta* is mentioned as a control but not shown in the Figure.

We thank Referee #3 for detecting this typo. We have now corrected it.

2. Figure 1F: No loading control is included.

We acknowledge Referee #3's comments. However, our aim was not to measure Ccq1 protein levels, but to measure the fraction of hyperphosphorylated Ccq1 in the absence of Ssu72. Our *loading control* for hyperphosphorylated Ccq1 is actually Ccq1. This technique was previously used in our lab (Carneiro et al., 2010 Nature) or other laboratories (e.g. Chang et al., 2013 PLoS Genetics) to measure protein phosphorylation levels.

3. Line 168: The superscript CTC1 with Cdc13 does not seem appropriate, as functional conservation between Cdc13 and CTC1 is not yet established.

Following Referee #3's suggestion, this superscript was deleted

4. Figure 3A: Untagged control was only included for one timepoint and not across the timecourse.

Untagged strains were used in every experiment in order to control for antibody specificity and experimental quality of our data. Mainly for technical reasons, we discarded replicas of untagged samples in cell cycle experiments as it significantly simplified our experimental set up. We would also like to point out that this experiment was repeated 4 times and our Stn1 cell cycle association to telomeres is equivalent with previous data published by the Nakamura's laboratory (Chang et al., 2013 Plos Genet).

5. Line 331-332: The sentence needs a qualifier (e.g. We wondered whether the observed telomere elongation...)

We thank Referee #3 for detecting this typo. It has now been corrected

I hope you had a good start into the new year, and thank you for submitting your revised manuscript for our consideration. Two of the original referees have now already assessed it, and I am happy to inform you that both are largely satisfied with the revisions and improvements to the paper. Referee 2 still retains a few minor reservations (see below), which I would invite you to answer/clarify during a final round of minor revision.

REFEREE REPORTS

Referee #2:

I think that this is an important and well-executed study. The revised manuscript is significantly improved. A few sticking issues are listed below.

Major points.

- I remain skeptical about the evidence in support of *cdc2* being the direct kinase affecting S74. In Fig EV3B, treatment with 1NM-PP1 might affect Stn1 recruitment indirectly, by altering cell cycle stage of the cells. I would find a control experiment (with DMSO and 1NM-PP1 treatment) on a *stn1-myc cdc2-as-M17* strain to be essential here.

- The legend to Fig 5B states 'HT1080 cells infected for 4 weeks'. This does not make sense and needs to be clarified. I presume it means 'infected, then grown out for four weeks'. It needs to be clarified when (and if) the SSU72 RNA was measured during the experiment (and also whether the STN1 RNA was measured). I am overall not too impressed by epistasis analysis under these conditions, with two knock-down 'alleles' and one phenotype in the single mutant which is quite mild. The efficacy of the knock-downs throughout the experiment should be monitored.

- Can the authors comment on the possibility that the reduced STN1 ChIP signal in Fig 6B might be confounded by cell cycle (indirect) effect of knocking down SSU72?

Minor points.

Why '(C)ST' instead of 'CST'? I mean, I understand why, but I do not see what this adds.

Line 34. change predominantly to predominant

Line 175. I actually do not understand what is meant by there being 'crosstalk' between the two mechanisms. There are here two pathways to inhibit telomere elongation, and removing both leads to neither additive nor multiplicative effects. In the absence of predictive models on how the release of either fully independent or inter-dependent repressive pathways would affect telomere length quantitatively, I am unable to extract any information from the fact that telomere length in the double mutant is greater than the sum in either single mutant.

Line 195-196. 'In contrast to our previous genetic studies' is confusing and should be clarified or eliminated.

Line 269. It is a strain that is constructed, not an 'allele'.

Referee #3:

The revised version of the manuscript has improved the quality and significance of the overall study and addressed many of my concerns. While the reviewers were not able to address several of my

concerns regarding the conservation of the SSU72 regulation of STN1 in humans cells, their additional data demonstrating that combined knockdown of STN1 and SSU72 give an epistatic-like effect and that SSU72 knockdown increased G-overhang length provide enough additional evidence to recommend that their manuscript be accepted for publication. Their findings provide an important contribution to understanding telomere length regulation.

Answers to Reviewers Comments:

Referee #2:

I think that this is an important and well-executed study. The revised manuscript is significantly improved. A few sticking issues are listed below.

Major points.

- I remain skeptical about the evidence in support of cdc2 being the direct kinase affecting S74. In Fig EV3B, treatment with 1NM-PP1 might affect Stn1 recruitment indirectly, by altering cell cycle stage of the cells. I would find a control experiment (with DMSO and 1NM-PP1 treatment) on a *stn1-myc cdc2-as-M17* strain to be essential here.

We agree with the referee that we provide no evidence that Cdc2 directly phosphorylates Stn1 at residue S74. In fact, we have stated as much in our rebuttal letter: "Nevertheless, we duly acknowledge that our result may reflect an indirect effect of Cdc2 on Stn1."

To make this point completely clear, we have added the following sentences to our manuscript hoping this would convey a fairer reflection of our uncertainty: In the Results section: "(...)our data indicates that *Cdc2^{Cdk1}* activity counteracts *Ssu72* phosphatase, perhaps indirectly via cell cycle regulation." and in Discussion: "However, we cannot rule out that the rescue of *Stn1* recruitment to telomeres achieved by *Cdc2* inactivation in *ssu72Δ* cells is indirect, perhaps working via a cell cycle dependent event."

- The legend to Fig 5B states 'HT1080 cells infected for 4 weeks'. This does not make sense and needs to be clarified. I presume it means 'infected, then grown out for four weeks'.

We thank the referee for detecting this typo. We corrected it in the current version of our manuscript.

It needs to be clarified when (and if) the SSU72 RNA was measured during the experiment (and also whether the STN1 RNA was measured).

RNA levels were monitored 4 weeks after infection. We have now modified the text in Figure legend Figure 5B and in Materials and Methods to reflect this.

I am overall not too impressed by epistasis analysis under these conditions, with two knock-down 'alleles' and one phenotype in the single mutant which is quite mild. The efficacy of the knock-downs throughout the experiment should be monitored.

We agree with the reviewer that SSU72 shRNA knockdown produces a mild increase of median telomere length of 12-14%, as monitored by TRF analysis. However, this elongation

was consistently observed in multiple experiments including the ones shown in Figure 5A and the previous rebuttal letter (Explanatory Figures 8, 9 and 10). In addition, our KDown effect of STN1 in these experiments is comparable to the ones performed by Chen et al. *Nature* 2012 after one month of infection, consistent with having used the same shRNA target sequence.

We monitored the efficiency of SSU72 knockdown by RT-PCR and of STN1 by WB at 4 weeks after infection with expected results (Explanatory Figure 1A). The experiment presented in Figure 5B was repeated twice during the 3-month revision process with similar results.

- Can the authors comment on the possibility that the reduced STN1 ChIP signal in Fig 6B might be confounded by cell cycle (indirect) effect of knocking down SSU72?

We have monitored now the cell cycle profile of SSU72 KDown cells after 4 months of infection and compared them with control Luciferase KDown. As shown for fission yeast, even though DNA damage is present in *ssu72* deficient cells, cell cycle profiles are not obviously affected (Explanatory Figure 1B). This result indicates that Stn1 recruitment defect observed in absence of Ssu72 function are unlikely to be attributed to a gross cell cycle defect.

Minor points.

Why '(C)ST' instead of 'CST'? I mean, I understand why, but I do not see what this adds.

To date, no CTC1 homologue has been identified in fission yeast. In order to differentiate the human and budding yeast CST complexes with the fission yeast Stn1-Ten1 complex, we used (C)ST, as previously used by other authors (e.g. Chang et al., *Plos Genetics* 2013), to allow for the potential conservation of an yet unidentified homologue.

Line 34. change predominantly to predominant

Thank you for detecting this typo that has been corrected.

Line 175. I actually do not understand what is meant by there being 'crosstalk' between the two mechanisms. There are here two pathways to inhibit telomere elongation, and removing both leads to neither additive nor multiplicative effects. In the absence of predictive models on how the release of either fully independent or inter-dependent repressive pathways would affect telomere length quantitatively, I am unable to extract any information from the fact that telomere length in the double mutant is greater than the sum in either single mutant.

We agree with the referee that the only secure information we can extract from our telomere length analysis is that *rif1*Δ and *ssu72*Δ are not epistatic. However, as pointed out by Referee #3, telomere length of *rif1*Δ *ssu72*Δ double mutants are longer than the additive effect of single mutants, suggestive of a synergistic effect. One could foresee that disruption of origin firing (under the control of Rif1) and lagging strand synthesis (regulated by Ssu72) may interact to produce exacerbated effects. However, further experiments are clearly needed to test this genetic assertion.

Line 195-196. 'In contrast to our previous genetic studies' is confusing and should be clarified or eliminated.

The phrase has been corrected to “*In contrast to our previous genetic studies presented in Fig 2C, (...)*”

Line 269. It is a strain that is constructed, not an 'allele'.

The term “allele” has been deleted from the sentence.

Referee #3:

The revised version of the manuscript has improved the quality and significance of the overall study and addressed many of my concerns. While the reviewers were not able to address several of my concerns regarding the conservation of the SSU72 regulation of STN1 in humans cells, their additional data demonstrating that combined knockdown of STN1 and SSU72 give an epistatic-like effect and that SSU72 knockdown increased G-overhang length provide enough additional evidence to recommend that their manuscript be accepted for publication. Their findings provide an important contribution to understanding telomere length regulation.

We thank Referee #3 for the suggestions that have improved our manuscript.

A)

B)

	Luciferase shRNA	SSU72 shRNA
G ₁	75.18	75.03
S phase	14.18	11.11
G ₂ /M	10.64	13.86

Explanatory Figure 1: A) WB analysis of STN1 and RT-qPCR for SSU72 mRNA expression related to the experiment presented in figure 5B. Samples were collected 4 weeks after infection. B) Cell cycle profile of SSU72 shRNA cells is similar to Luciferase control. FACS analysis DNA collected from HT1080 cells infected for 4 weeks with lentiviral particles carrying shRNAs against SSU72 (UTR regions) and Luciferase (Luc).

YOU MUST COMPLETE ALL CELLS WITH A PINK BACKGROUND ↓
PLEASE NOTE THAT THIS CHECKLIST WILL BE PUBLISHED ALONGSIDE YOUR PAPER

Corresponding Author Name: Miguel Godinho Ferreira
Journal Submitted to: Nature Structural and Molecular Biology
Manuscript Number: EMBOJ-100476